# Arctic sea-ice ridges are biomass hotspots harboring diverse microbial communities

Oliver Müller [1,22] ✉, Jessie Gardner[2,22], Lasse Mork Olsen [1,3], Evgenii Salganik [4,5], Philipp Assmy [4], Rolf Gradinger [2], Gunnar Bratbak [1], Clara J. M. Hoppe [5], Benjamin A. Lange [4,6], Morven Muilwijk [4], Dmitry V. Divine [4], Nicole Aberle[7,8], Jeffrey W. Krause [9,10], Marit Reigstad [2], Eva Leu [2,11], Tatiana M. Tsagaraki[1], Aud Larsen [12], Knut V. Høyland[13], John Paul Balmonte [14,15], William Boulton[16], Håkon Dahle[1], Lena Eggers[5], Allison A. Fong[5], Gaël Guillou[17], Benoit Lebreton[17], Katja Metfies [5], Thomas Mock [18], Elzbieta Petelenz[1], Agnieszka Tatarek[19], Sinhué Torres-Valdés [5], Anders Torstensson [20,21], Jozef Wiktor[19] & Mats A. Granskog [4]

Although sea-ice ridges are prominent features of the Arctic Ocean, very little is known about their role as habitats and in biogeochemical cycles. Here, we show that ridges provide complex sea-ice habitats which host unique and diverse biological communities. Seasonally, ridges appear to transition from a biological repository in winter to biological hotspots in summer, surpassing algal biomass in level ice and surface waters by up to eight-fold. In summer, ridges can contain up to 80% of the total area integrated sea-ice algal biomass, emphasizing their importance in the Arctic sea-ice ecosystem. However, environmental shifts, such as meltwater infiltration and freezing inside the ridge in late summer, alter microbial communities from being predominantly autotrophic to heterotrophic. Our work provides evidence of contrasting roles of sea-ice ridges for Arctic carbon cycling in summer and shows that the habitats in the ridge interior harbor unique microbial communities, adding complexity to Arctic biodiversity.

Pressure ridges are a characteristic feature of the Arctic ice pack. They occupy up to 50% areal fraction and a considerable portion of the sea-ice volume, though their contributions vary seasonally and regionally (e.g., refs. 1–3). Ridges usually form from 0.2–0.4 m thick ice blocks during deformation events when ice floes collide, and have a sail above and a keel below the water level[4]. Keels can protrude more than 40 m into the ocean (e.g., refs. 5,6), but are typically 4 to 10 m deep[4,7]. The keel is composed of randomly arranged ice blocks (rubble), with voids initially filled with seawater (typically with a fraction of 30%)[7].

Within the diverse habitats in sea ice, aquatic biota have evolved to exploit unique ecological niches, ranging from freshwater melt ponds on summer multi-year ice, the brine channel system within sea ice, to the bottom of the sea ice (e.g., refs. 8–11). These organisms form distinctive ice-associated biological communities. A larger diversity of habitats for Arctic biota can be found within the keel rubble compared to the bottom of level ice (e.g., refs. 8,12,13). These habitats include water-filled voids in the rubble (macroporosity), surfaces of the ice blocks facing different directions, and brine channels in the ice blocks (microporosity). Due to their greater ice thickness compared to level ice, ridges provide a habitat that can survive the summer melt when level ice gets thinner and eventually melts (Fig. 1).

Ridges have thus been suggested to provide a refuge for ice-associated organisms during the melt season, where organisms can also escape low-salinity surface meltwater layers[13].

Although ridges provide a range of different sea-ice habitats and compose a substantial part of the Arctic ice pack, biological information on this habitat is scarce. This knowledge gap is attributed to the difficulties in sampling ridges due to their great thickness and complex structure. Pioneering ecological studies by divers[14] provided evidence of accumulations of under-ice zooplankton and macrofauna as well as sea ice algae in association with pressure ridges. Syvertsen[8] additionally demonstrated that distinct diatom assemblages are associated with the underside and upper surfaces of submerged ridge blocks. More recent studies identified specific locations in pressure ridges as biological hotspots[15], as well as abundances of ice meio- and macrofauna exceeding that of level ice[8,12–14,16–18]. However, these studies were limited spatially and temporally, representing only single snapshots in the summer and samples from the exterior flanks of the ridge keels. In addition, research addressing microbial communities within sea ice ridges has been limited to using microscopy techniques with a focus on eukaryotes[15]. On the contrary, viruses[19,20], archaea and bacteria have been extensively studied in level ice[21–25], providing evidence that sea ice

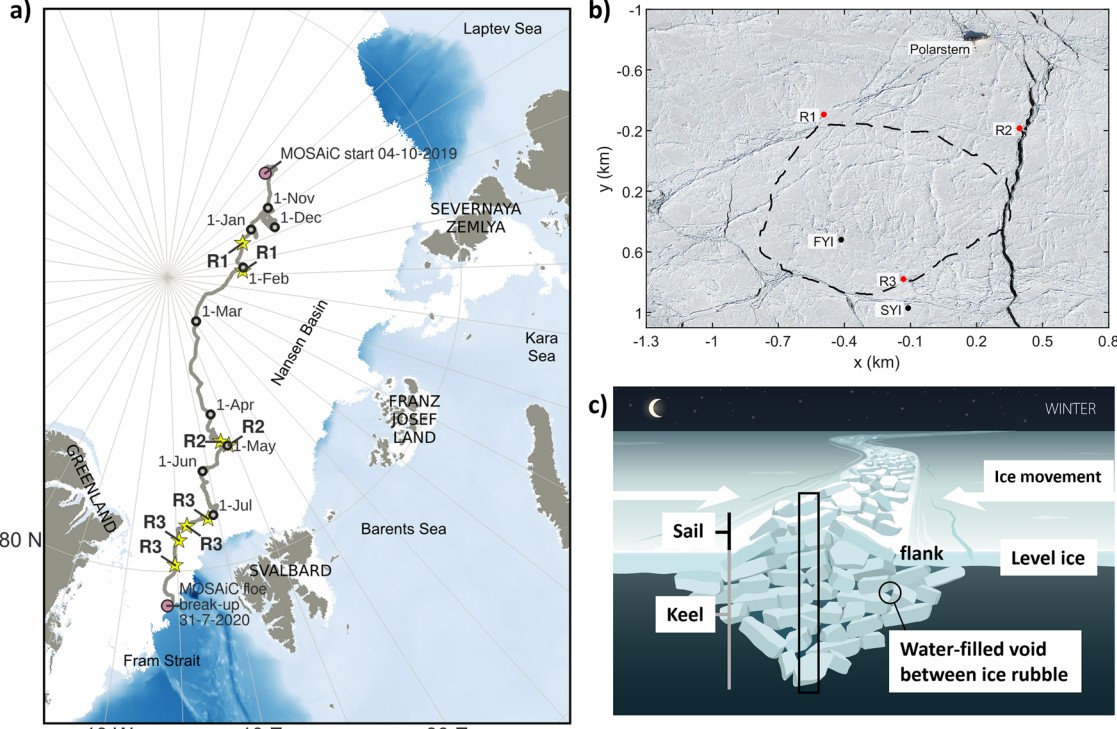

**Fig. 1 | MOSAiC drift and locations where ridges were sampled. a** Ridge-specific sampling locations as part of the MOSAiC drift (gray line) during winter (R1), spring (R2), and summer (R3) with ridge sampling events indicated by the yellow stars. The first day of each month is indicated with a black circle. The white color indicates the sea ice extent on 15 July 2020 from AMSR2[130], while the light blue to dark blue colors indicate bathymetry (IBCAO[131]). **b** Aerial photograph from 23 April 2020 showing the MOSAiC Central Observatory ice floe with the locations of the three ridges (R1, R2, and R3), the first-year ice (FYI) and second-year ice (SYI) coring sites, the RV

Polarstern, as well as the outline of the ice floe during summer (dashed line). Although all three ridges are visible on this image from April, they were not accessible throughout the whole drift since ice dynamics changed the conditions and accessibility at the MOSAiC Central Observatory over time (see "Methods"). **c** Conceptual visualization of a sea ice ridge during winter. The black frame displays where sea ice cores were taken (for more details, see "Methods" and Table 3). Credits: **b**: Niklas Neckel; **c**: Frida Cnossen.

serves as a seasonally dynamic habitat for diverse microbial communities. Bacteria and viruses can be found in high abundances in the bottom layers during periods of high biological productivity and within the interior of the level ice in layers with high biomass when brine channels facilitate connectivity and nutrient exchange. Abundant bacterial taxa found in level ice include members of the *Flavobacteriia* and *Gammaproteobacteria* in summer, while ammonia-oxidizing archaea are prevalent during winter[11,24]. Far less is known about the temporal and spatial distribution of all biota in different ridge habitats in different regions of the Arctic, their seasonal variation, and how they respond to varying environmental conditions.

The Multidisciplinary drifting Observatory for the Study of Arctic Climate (MOSAiC) expedition (Fig. 1) provided a unique opportunity to better understand the interlinked physical, chemical, and biological systems over a seasonal cycle in the central Arctic Ocean. This study examines the physical (e.g., refs. 26,27), biological, and biogeochemical characteristics of pressure ridges and, to the best of our knowledge, presents first measurements from the interior of the ridge keels, thereby providing a detailed ecological inventory of different ridge habitats and how they change from winter to summer, allowing us to address the following research questions: (1) How are habitats and associated communities in ridges changing from winter to summer; (2) Are those habitats and communities different from those of level ice and (3) Could ridges provide a refuge for ice-associated Arctic flora and fauna if level ice melts completely during the summer?

Our observations demonstrate that sea-ice pressure ridges harbor unique and complex habitats with diverse and biomass-rich biological communities, confirming their important role in the Arctic ecosystem. Despite a ridge areal fraction of only 22% of the total sea ice cover, we estimated that the habitable ice surface area in a highly consolidated ridge in mid-summer (July) is still 5–10 times greater compared to level ice, and the

interior maze of water-filled voids provides an additional habitat for organisms ranging from microbes to polar cod. We further estimated that ridges could contribute up to 80 ± 20% of chlorophyll *a* (Chl-a) per m² of Arctic sea ice. The structural complexity of ridge habitats contributes to the overall high variation in community composition, and the change from water-filled to frozen voids in summer adds to the emerging complexity and importance of ridge communities and processes. Our work highlights the need to better assess the contribution of ridges to the overall sea-ice-related biogeochemical cycling and biodiversity to be able to predict the consequences of the changing sea-ice conditions in the Arctic.

## Results
### Seasonal changes in the physical characteristics of ridge habitats

Physicochemical characteristics of the ridges showed substantial seasonal variations. These data come from different ridges (R1–3) at different times of the year due to logistical constraints imposed by sea-ice dynamics (see "Methods"). R1 formed in fall 2019 and had matured for 3–4 months before sampling in January 2020. Its average bulk ice salinity was about 4, and between 3 and 9 for the collected ice samples (Supplementary Fig. 1). From about 2 m keel-depth downward from the water-level, the ice was soft, slushy, and porous, indicating limited consolidation (freezing between ice blocks). Water-filled voids at about 1.9 and 2.0 m had a salinity (29.9; reported on the Practical Salinity Scale 1978, PSS-78, which is dimensionless) similar to surrounding seawater (31.8). The ice surrounding the void had brine volume fractions of 6–18% (Supplementary Fig. 1), all above the permeability threshold (>5%[28]). R2, sampled in spring, was a young ridge formed from thin first-year ice (FYI) in March[29]. Bulk ice salinity was 6.9 ± 1.4 in April and

**Fig. 2 | Principal component analysis (PCA) of environmental variables.** Environmental variables (bulk salinity, brine volume, temperature, concentration of ammonium ($NH_4$), nitrite/nitrate ($NO_2/NO_3$), phosphate ($PO_4$) and silicic acid ($Si(OH)_4$)) of ridge ice samples, separated into void roof ice (up-triangle), frozen void (diamond), void bottom ice (down-triangle), ice with visible algae inclusion close to frozen voids (star) and ridge bottom ice (hexagon). Ridge ice samples are from January (R1, blue) and July before (R3, black) and after (R3, pink) meltwater-driven consolidation.

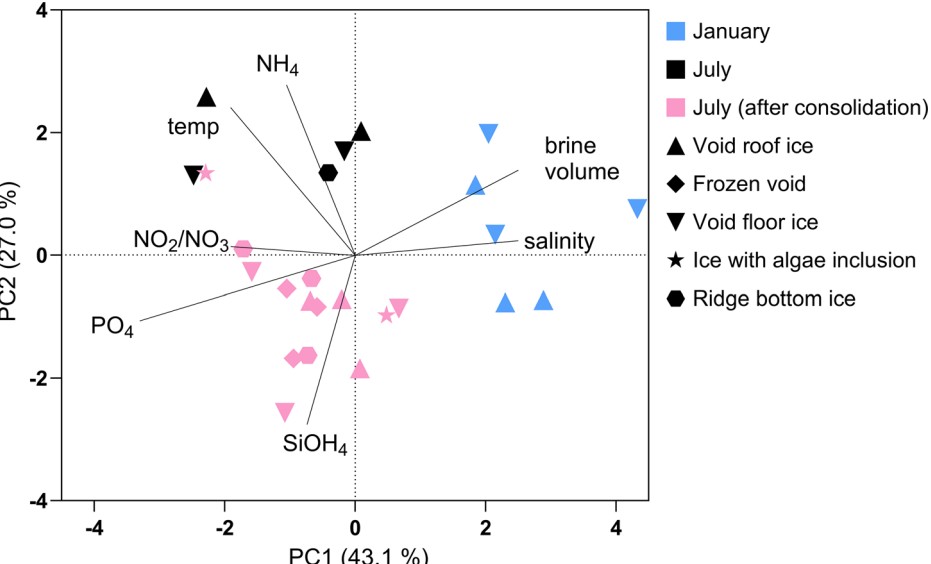

## Habitat complexity and associated microbial communities

Alongside seasonal differences in physicochemical ridge habitat properties (Fig. 2), Chl-a and particulate organic carbon (POC) concentrations as well as microorganism abundance and community composition differed temporally and spatially (Figs. 3 and 4). Generally, higher Chl-a concentrations were measured in the upper unconsolidated parts of the ridge keel with the highest Chl-a concentrations in ice samples above and below the upper water-filled void at the beginning of July (Fig. 3a: void roof ice: 35.5 and void floor ice: 60.1 mg Chl-a m$^{-3}$). In January and April, the highest Chl-a concentrations were also observed in ice block samples associated with voids, reaching 2–3 mg Chl-a m$^{-3}$. At the end of July, high Chl-a concentrations were encountered in ice block samples associated with a water-filled void at the ridge flank (void roof ice: 7.2 mg Chl-a m$^{-3}$). Concentrations in the deeper parts of the keel were generally lower (<5 mg Chl-a m$^{-3}$, Fig. 3a), except for some ice samples with visible inclusions of algal biomass. The overall highest Chl-a concentrations (Fig. 4) were measured in interior ridge ice samples and exceeded the highest values in level first-year (FYI) and second-year ice (SYI) (7.9 and 5.3 mg Chl-a m$^{-3}$, respectively), measured in bottom ice samples during spring and early summer, as well as in interior level ice core samples (5.3 and 4.0 mg Chl-a m$^{-3}$, respectively) in all seasons from winter to summer (Fig. 4a). Chl-a concentration in seawater (0.2–1.0 mg Chl-a m$^{-3}$) was 5–60 times lower than those measured in level ice and ridge ice, even during summer. Concentrations of POC inside the ridge were variable, with similar trends to those of Chl-a. High POC concentrations were measured in ice block samples associated with the upper void, both when voids were water-filled and frozen (Fig. 3b: void roof ice: 800 and 1350 mg C m$^{-3}$). In level FYI, the highest POC concentrations were measured in the bottom ice sections in July (up to 1590 mg C m$^{-3}$). The overall highest POC values were measured in level SYI in the interior ice sections at 80 to 100 cm depth, varying between 1540 and 3300 mg C m$^{-3}$.

We estimated volume fractions of ridged ice, level FYI and level SYI for a 40 km$^2$ area around the MOSAiC ice floe to compare Chl-a and POC standing stocks in the different ice types based on July data (see "Methods"; Tables 3 and 4). When integrating for the entire sea ice volume within this area, we estimated that ridges (25 ± 5 mg Chl-a m$^{-2}$) contained 80 ± 20% of the entire Chl-a standing stock when using the observed ridge areal fraction of 22%. FYI and SYI (both 1.4 ± 0.2 mg Chl-a m$^{-2}$), contributing 78% to the total sea ice cover, contained 17 ± 2% of the entire Chl-a standing stock. Likewise, we found that the ridges (1570 ± 140 mg C m$^{-2}$) contain more POC than FYI (440 ± 30 mg C m$^{-2}$) and SYI (1030 ± 90 mg C m$^{-2}$) combined.

5.9 ± 1.7 in May. Near voids at about 1.3–1.5 m depth, the ice temperature was close to the seawater freezing point, with brine volumes ≥10% (Supplementary Fig. 1). The voids' high location in the keel suggests limited consolidation, in line with the young age of the ridge. Although void salinity was not measured, the temperature of the adjacent ice was −2 to −3 °C, implying a salinity similar to seawater. R3, sampled from early to late summer, formed in February and had consolidated by late winter, with keel melting and ice temperatures close to seawater freezing point by July. Bulk ice salinity was 2.1 ± 1.2 for the full depth and ranging from 0.6–5.8 in 10-cm sections (Supplementary Fig. 1). On 3 July, void salinity varied vertically: 7.9 at 2.0–2.1 m (similar to under-ice meltwater[30];) and 29.5 at 3.0–3.8 m (close to seawater at 31.1). Brine volume fractions of the ice surrounding the water-filled voids ranged from 4–28%, and were mostly above 10% (Supplementary Fig. 1). After a melt pond drainage event between 3–10 July[31], ridge ice bulk salinity and brine volume fractions decreased. Some ice samples, representing frozen voids, had the lowest observed brine volume fractions of all sampling periods (<5%, Supplementary Fig. 1) and very low salinity (0.6–1.1[26]) with much less opacity compared to typical sea ice (Supplementary Fig. 2).

Changes in the dissolved nutrient concentrations of ridge ice and void water samples over time reflected changes in the under-ice water as the floe drifted through different water masses during the MOSAiC drift. In January, when the ice floe was in the Amundsen Basin, nitrate/nitrite concentrations in ridge ice and void water were low (<2 μM) and silicic acid slightly higher (up to 4 μM). By April/May the floe had drifted over the Gakkel Ridge into the Nansen Basin, accompanied by an increase in nitrate/nitrite (to 4–6 μM) and a decrease in silicic acid concentrations (to 2 μM) in the water-filled voids (Supplementary Fig. 3). This persisted into mid-July, after which nitrate/nitrite concentration decreased (<2 μM) and silicic acid increased (5–6 μM) in ridge ice, void water and under-ice water samples. Phosphate concentrations were lower than 0.5 μM and displayed a similar pattern as nitrate/nitrite (Supplementary Fig. 3). The relationship between the physical conditions and nutrient concentrations of ridge ice samples (Fig. 2; R2 not included due to missing nutrient data) highlights a distinct separation based on time of sampling. Samples from January are distinguished from the July samples by high bulk salinity and high brine volume fractions, whereas the samples from July cluster on the opposite end of PC1, correlated with low bulk salinity. They are grouped into two subclusters, before and after void freezing, due to differences in brine volume fraction (lower after the freezing) and temperature (higher before the freezing).

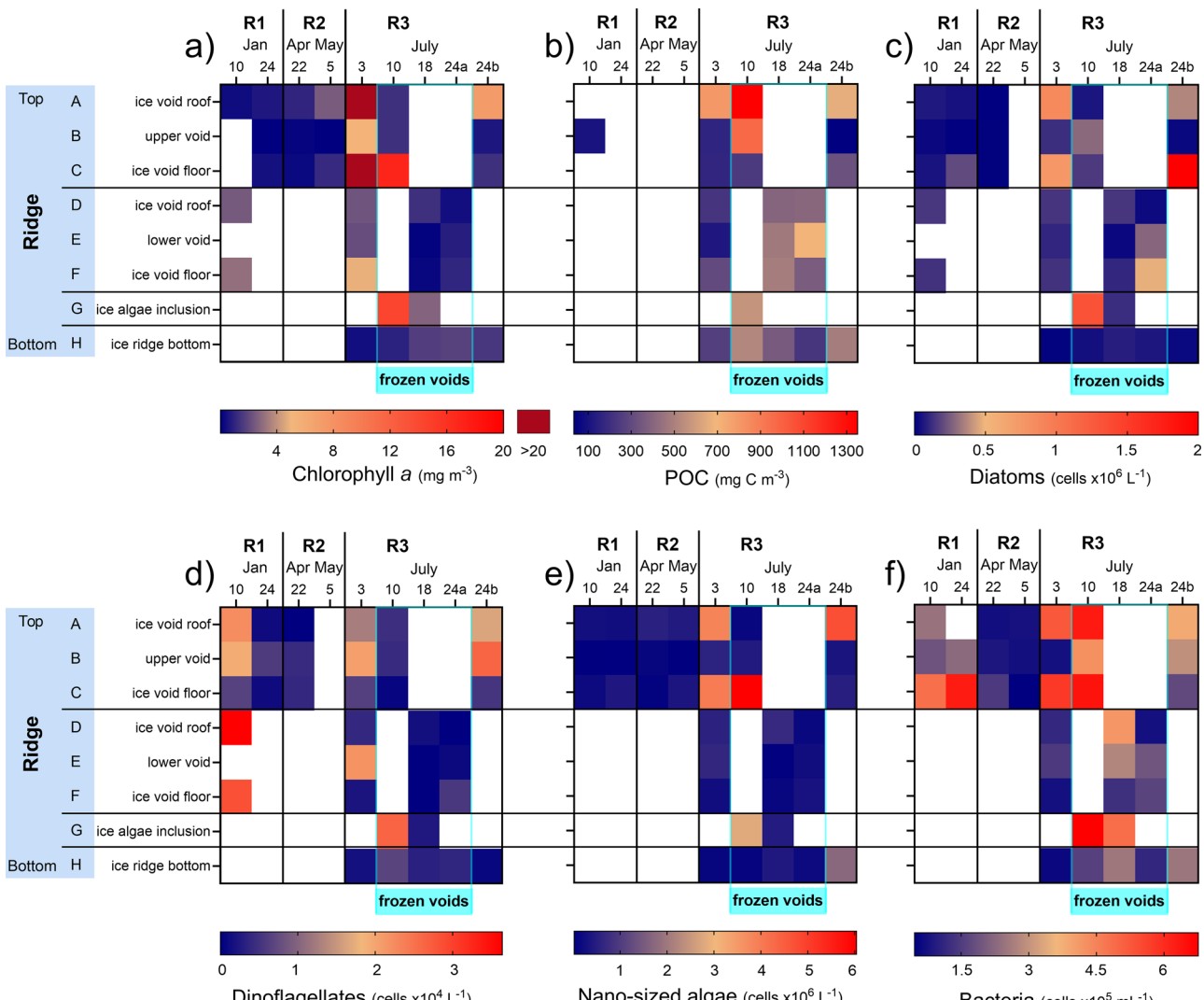

**Fig. 3 | Biomass and microbial abundance in different parts of the ridges through different seasons.** Changes in **a** chlorophyll *a* and **b** particulate organic carbon (POC) concentrations and abundances of **c** diatoms, **d** dinoflagellates, **e** nano-sized algae and **f** bacteria in samples from different ridges and sampled in different seasons (letters A–H indicate the sample location as illustrated in Fig. 10). White background indicates no data and the light blue frame (frozen voids) indicates the period when water filled voids were frozen.

Abundances of protists, including diatoms, dinoflagellates, and nano-sized (2–20 μm) algae (Fig. 3c–e) varied throughout the ridge and between seasons. Diatoms and nano-sized algae resembled Chl-a concentration patterns, with the highest abundance measured in July. Independent of the group-specific differences, the highest concentrations for all three groups (diatoms: $2 \times 10^6$ cells $L^{-1}$; nano-sized algae: $6 \times 10^6$ cells $L^{-1}$; dinoflagellates: $3.6 \times 10^4$ cells $L^{-1}$) were detected in ice samples around water-filled voids. Dinoflagellates dominated the protist community in ridge keel samples in January (Fig. 3d: $2$–$3.6 \times 10^4$ cells $L^{-1}$), and were also abundant in summer, predominantly in water-filled voids or the adjacent ice. Their abundances were greatly reduced in frozen voids.

Bacterial abundances were highest in July in the frozen voids (up to $6.4 \times 10^5$ cells $mL^{-1}$), but higher bacterial abundances were also measured in void floor ice samples in January (Fig. 3f). Changes in bacterial abundance (Fig. 4b) were comparable to the pattern observed for Chl-a concentrations (Spearman's $\rho = 0.3$, $p < 0.0001$; $n = 307$), with high abundances in samples with high Chl-a, but differences in maximum bacterial abundance values, ranging between $0.6 \times 10^6$ (seawater) and $1.1 \times 10^6$ cells $mL^{-1}$ (interior FYI, June 22; Supplementary Table 1), were less pronounced between environments than for Chl-a. Interestingly, the highest bacterial abundance in ridge bottom ice ($2.5 \times 10^5$ cells $mL^{-1}$) was two to three times lower than bottom

SYI ($5.3 \times 10^5$ cells $mL^{-1}$) and FYI ($7.0 \times 10^5$ cells $mL^{-1}$), respectively. There was a general increase in bacterial abundances from winter to summer in FYI and SYI bottom ice, which was most pronounced in FYI bottom ice during the spring to summer transition (Fig. 4b). This increase was also seen in seawater, with the highest abundances in the second half of July in the upper 20 meters ($0.6 \times 10^5$ cells $mL^{-1}$).

Ordination analyses of prokaryotic (16S rRNA gene) and eukaryotic (18S rRNA gene) sequence data to identify beta diversity patterns also revealed spatial and temporal differences in community compositions based on the Bray-Curtis dissimilarity index (Non-metric Multidimensional Scaling—NMDS; Fig. 5a, b). Both the pro- and eukaryotic communities separate into four main clusters, with significant differences in community composition (Supplementary Table 2). The clusters are separated by season and sample type (winter/spring vs. summer, and under-ice water vs. ridge ice) and the state of the voids (water-filled vs. frozen). The prokaryotic community composition clusters separately for both water-filled void and under-ice water samples from winter/spring and summer (Fig. 5a; purple and turquoise background). A SIMPER (Similarity Percentage) analysis showed that *Polaribacter* contributed most to the summer, while *Nitrosopumilus* and *Pelagibacter* contributed most to the winter cluster. Bacterial communities observed in summer in the upper water-filled void with low

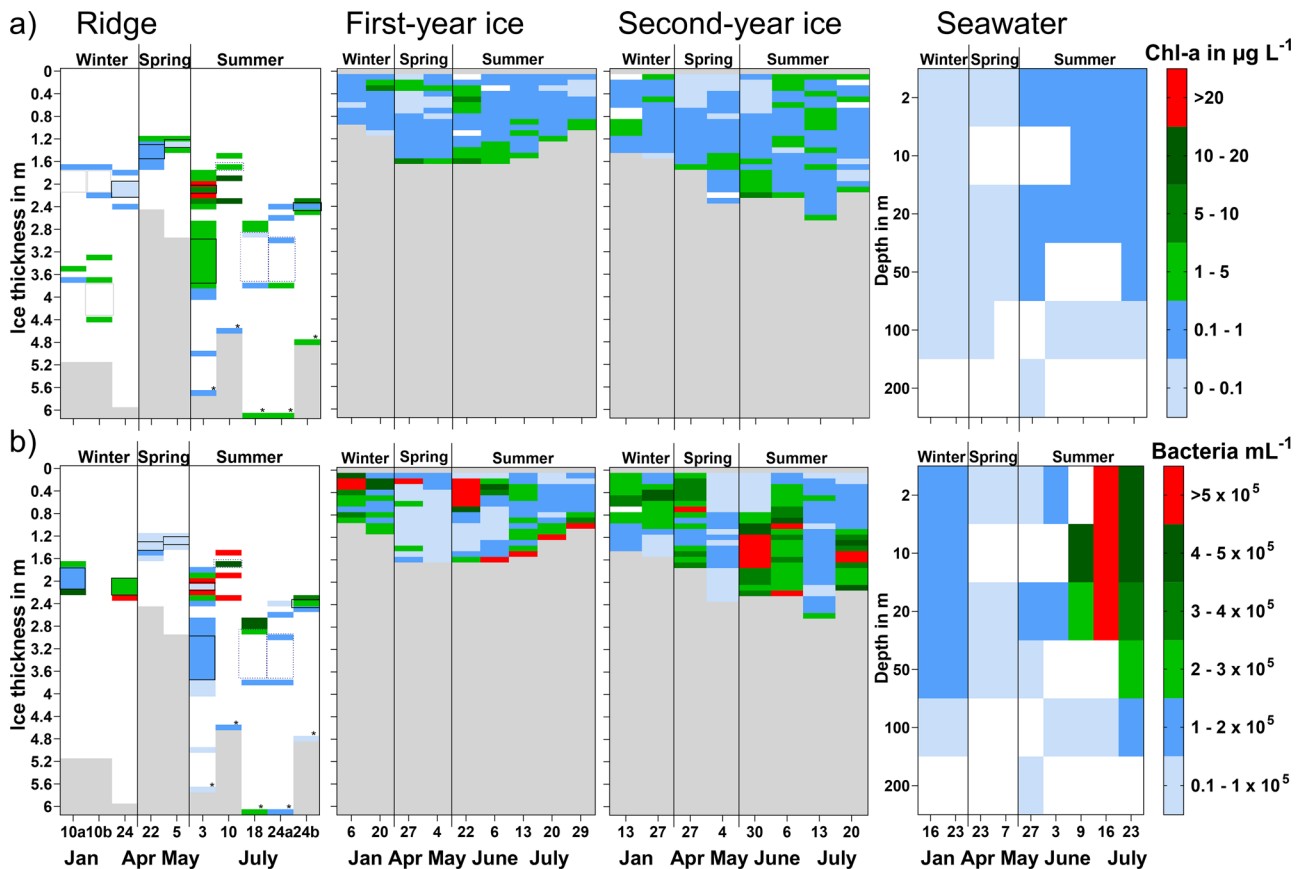

**Fig. 4 | Chlorophyll *a* concentration and bacterial abundance in ridge keels, level ice and seawater through different seasons. a** Chlorophyll *a* (Chl-a) concentrations and **b** bacterial abundance in ridge keels, first-year ice (FYI), second-year ice (SYI) and seawater (<200 m). White background indicates no data and gray background indicates sea water below the ice. The asterisk indicates ridge bottom ice samples and the frames the extent of water-filled (continuous) or frozen (dotted) voids in the ridge. For the type of ridge samples shown, see details in Fig. 3a, f.

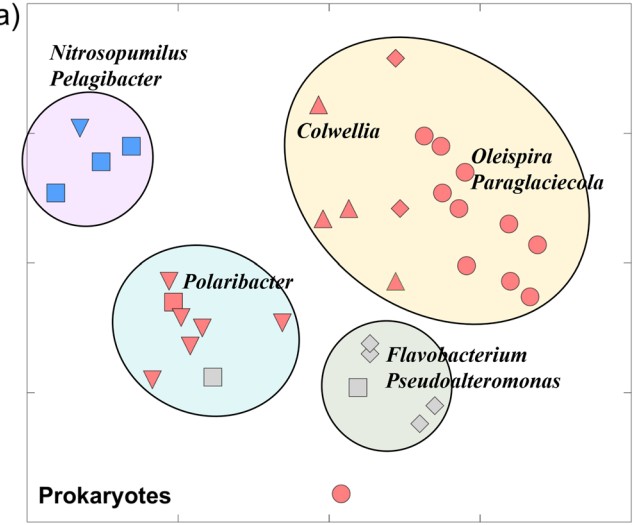

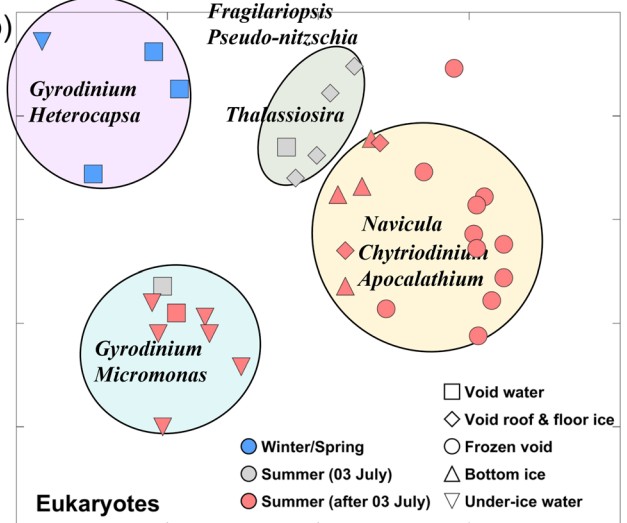

**Fig. 5 | Ordination analysis of prokaryotic and eukaryotic community composition.** Non-metric multidimensional scaling analysis of **a** the prokaryotic (16S rRNA gene) and **b** the eukaryotic (18S rRNA gene) community composition (stress value = 0.11). The sequence data were square-root transformed and sample similarity calculated using the Bray–Curtis dissimilarity index. Samples are visualized based on season (color) and sample location (shape). Four prokaryotic (**a**) and eukaryotic (**b**) clusters are highlighted with a green, yellow, turquoise, and purple background. The sequences that contributed most to the observed grouping based on a SIMPER analysis are visualized with genus names.

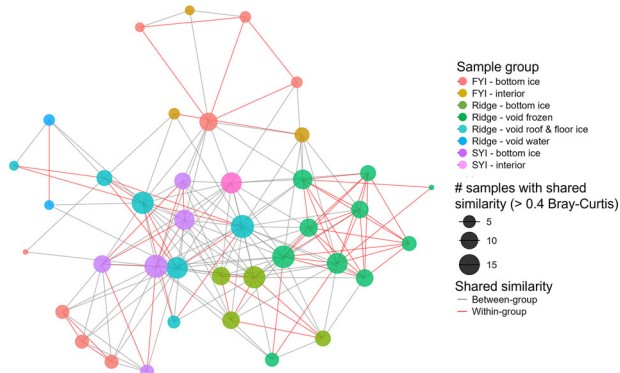

**Fig. 6 | Network diagram illustrating similarity of the eukaryotic community composition across ridge, first- and second-year ice.** Undirected network based on sequence data of the eukaryotic (18S rRNA gene) community composition in ridge, first-year ice (FYI) and second-year ice (SYI) showing the similarity in community composition between different samples and environments (Bray–Curtis similarity threshold >0.4). Community similarities within groups are visualized using red lines and between groups using gray lines. The size of the node indicates the overall numbers of samples with shared similarity. Winter samples formed an outgroup, highly separated from the spring/summer cluster, and were therefore excluded from the analysis.

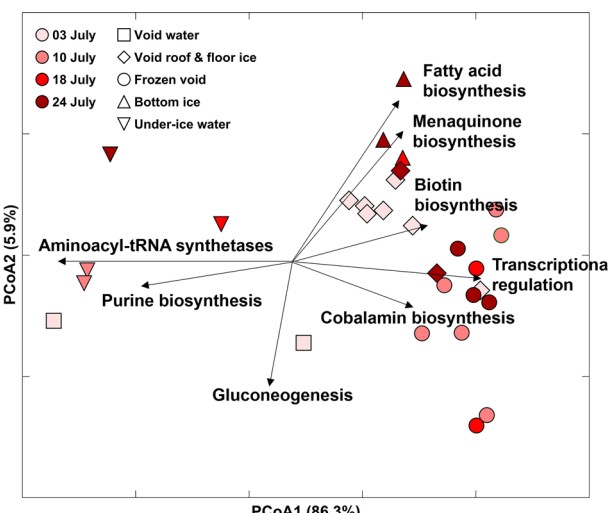

**Fig. 7 | Principal coordinates analysis (PCoA) of gene abundance in ridge samples.** PCoA of gene abundance data from metagenome sequencing assigned to COG (clusters of orthologous groups) pathways. Samples are visualized by time of sampling (color) and different locations (shape). The COG pathways that contributed most to the observed grouping are visualized with a black arrow and simplified pathway names.

salinity and the ice samples connected to the water-filled voids from 3 July were very similar and with high relative abundances of *Flavobacterium* and *Pseudoalteromonas* (Fig. 5a, green background). The 16S rRNA gene sequences were further grouped into two sub-groups for ridge ice samples after 3 July, one group with a high abundance of *Colwellia*, *Oleispira*, and *Paraglaciecola*, mainly in samples from the frozen void, and the ridge bottom-ice samples dominated by another *Colwellia* ASV (amplicon sequence variant) (Fig. 5a, yellow background). The analysis of eukaryotic community composition showed a similar grouping, and a Mantel test indicated a strong correlation between the eukaryotic and prokaryotic community composition (Mantel's $r = 0.769$; $p = 0.001$). There were two clusters of samples from the lower void and under-ice water separated by either high *Gyrodinium* and *Heterocapsa* abundance in winter or high *Micromonas* and *Gyrodinium* abundance in summer (Fig. 5b; purple and turquoise background). There was another cluster, separated due to high abundances of different diatom genera (*Fragilariopsis*, *Pseudo-nitzschia*, and *Thalassiosira*), consisting mainly of void ice and void water samples from 3 July (Fig. 5b, green background). Ridge ice samples collected after 3 July clustered separately and had high abundances of *Navicula*, and the dinoflagellates *Chytriodinium* and *Apocalathium* (Fig. 5b, yellow background).

The eukaryotic community composition in ridge samples differed significantly from those in FYI and SYI independently of the state of the ridge (water-filled—frozen voids, Supplementary Table 2 and Supplementary Fig. 4). Alpha diversity indices, including Chao1 and Shannon, were comparable across the environments and indicate similar within-habitat richness and evenness (Supplementary Fig. 5a, b). The indices showed large differences within sample groups, indicating large variation in taxonomic diversity between samples in the three groups. Interestingly, gamma diversity, expressed as the number of unique genera per environment, showed a higher diversity in the ridge than in FYI and SYI samples (Supplementary Fig. 5c, d), with 12.2% of a total of 442 genera unique to the ridge samples, while 9.3% and 8.1% were unique to FYI and SYI, respectively. The higher gamma diversity in ridge samples was mainly due to unique diatom and ciliate taxa (Supplementary Fig. 5d). This difference in gamma diversity, together with the clear differences in beta diversity (Supplementary Table 2), indicates that ridge habitats host distinct protist communities. The largest differences in community composition between ridge and level ice were observed for ridge samples taken after 3 July, when the voids were frozen (Supplementary Fig. 4). This distinction is also supported by the network

analysis that identified taxa relationships between ice types (ridge; FYI; SYI; Fig. 6) demonstrating community similarities within distinct environmental categories (e.g., frozen void, SYI- and FYI-bottom ice, void roof & floor ice), despite an overall high degree of similarity between habitats. Ridge samples collected after 3 July (Fig. 6: Ridge—void frozen) showed the strongest within-group similarity (red lines in Fig. 6). FYI groups are more dispersed, with moderate within-group similarity and more similarity shared between other groups. Both FYI and SYI samples showed more similarity between groups than within their own groups. SYI-bottom and interior ice samples as well as ridge void roof and floor ice showed the largest nodes, indicating a shared similarity with a large number of other samples.

The analysis of relative gene abundance data from metagenome sequencing assigned to COG (clusters of orthologous groups[32]) pathways showed functional differences between under-ice water, including the lower water-filled void, and the ridge ice and upper water-filled void sampled in July (Fig. 7). The under-ice water samples grouped together due to a higher abundance of genes of the aminoacyl-tRNA synthetase and purine biosynthesis pathways. These genes code for proteins that are involved in protein synthesis, cell viability, and energy metabolism, and might be important in conditions with varying nutrient concentrations[33,34]. Ridge ice samples showed a higher abundance of genes coding for enzymes involved in transcriptional regulation and cobalamin biosynthesis, which are indicative of fast-growing bacteria and algae-growth-promoting potential, respectively[35,36].

Genes encoding enzymes involved in algae-growth promoting cofactor pathways, such as cobalamin (KEGG module ID: M00122) and thiamine (M00899) synthesis, were also relatively more abundant in metagenome assembled genomes (MAGs) from interior ridge ice samples (19% and 7%) compared to MAGs from ridge bottom ice samples (7% and 1%) (Supplementary Table 3). The higher abundance of bacteria in ridge ice samples when the voids were frozen (Fig. 3f) was also reflected in the relative contribution of high nucleic acid-containing (HNA) bacteria, an indicator of fast-growing bacteria. HNA bacteria accounted for up to 89% of the total bacterial abundance in frozen void samples and were overall higher when compared to water-filled voids (Mann–Whitney $U$ test; $p = 0.02$). Moreover, bacterial production (up to 24.2 mg C m$^{-3}$ d$^{-1}$) was significantly higher in frozen void samples compared to ridge ice samples collected when the voids were water-filled (Mann–Whitney $U$ test; $p = 0.03$; Fig. 4 and Supplementary Fig. 6).

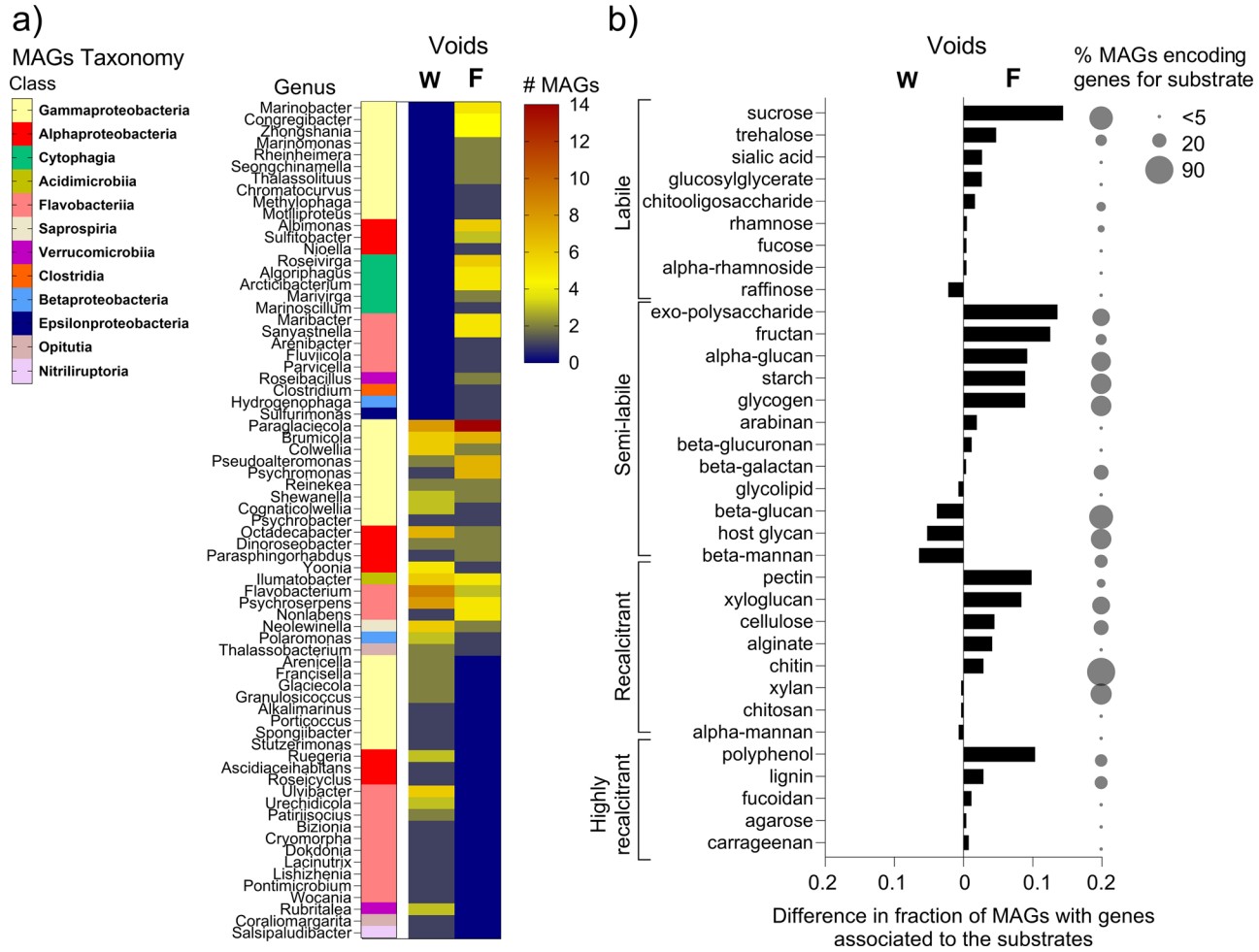

**Fig. 8 | Distribution of bacterial metagenome-assembled genomes and their potential for carbohydrate degradation in ridge ice samples when voids were water-filled and frozen. a** Heatmap of metagenome-assembled genomes (MAGs) with assigned taxonomy in ridge ice samples collected when voids were water-filled (W; $n = 122$) and when they were frozen (F; $n = 143$). Taxonomic assignments are based on the best-matching reference genome with the highest ANI score and are presented at the genus level. Class level is indicated by the color code and MAGs without assigned taxonomy were excluded. **b** Changes in the potential for carbohydrate degradation based on functional gene annotations derived from dbCAN using the Carbohydrate-Active Enzyme (CAZy) database using MAGs from ridge ice samples collected when voids were water-filled (W; $n = 270$) and when they were frozen (F; $n = 266$). The black bars show the difference in the fraction of MAGs per habitat (not frozen vs. frozen voids) containing genes involved in the processing of different carbon substrates grouped by recalcitrance. The bubble size indicates the overall percentage of MAGs encoding genes for each substrate.

The taxonomic assignments of MAGs showed a higher abundance of taxa belonging to Gammaproteobacteria and Bacteroidota (including Flavobacteriia and Cytophagia) in ridge ice samples collected when voids were frozen (Fig. 8a). The relative proportion of MAGs assigned to Gammaproteobacteria taxa was almost twice as high in frozen voids (27%) compared to samples of ridge ice surrounding water-filled voids (16%). MAGs representing genera belonging to the class of Cytophagia (*Roseivirga*, *Algoriphagus*, *Arcticibacterium*, *Marivirga*, and *Marinoscillum*), as well as specific genera from the other classes, including *Albimonas* and *Sulfitobacter* belonging to Alphaproteobacteria, were only present in ridge ice samples when voids were frozen. Overall, 50% of MAGs were unique to samples from ridge ice when voids were frozen, compared to 33% in samples from ridge ice surrounding water-filled voids. The analysis of metabolic functions showed that genes encoding proteins involved in amino acid utilization (various aminotransferases), aromatics degradation (ubiX, catA), fermentation (adh, ldh, pflD), complex carbon degradation (pullulanase, glucoamylase) and nitrogen cycling (norBC, napAB, nosDZ, nirBDK) were relatively more abundant in MAGs coming from ridge ice samples when voids were frozen compared to ridge ice samples surrounding water-filled voids (Supplementary Table 4). We further investigated the potential for carbohydrate degradation of the MAGs based on functional gene

annotations derived from dbCAN using the Carbohydrate-Active Enzyme (CAZy) database with focus on glycoside hydrolases, polysaccharide lyases, carbohydrate esterases, carbohydrate-binding modules and auxiliary activities (Fig. 8b). Overall, genes encoding enzymes involved in the processing of carbon substrates of different complexity were more prevalent in ridge ice samples when voids were frozen (26 substrates) than when they were water-filled (8 substrates). In particular, genes associated with the processing of sucrose, exopolysaccharide, fructan, glycogen, starch, pectin, xyloglucan and polyphenol were found in more MAGs that were retrieved from ridge ice samples when voids were frozen (Fig. 8b and Supplementary Table 5).

## Discussion

### Ridges provide unique habitats for sea ice-associated biota

Superimposed on seasonal changes in the Arctic icescape, the complex three-dimensional structure of ridges, combined with strong vertical and horizontal environmental gradients, facilitates large habitat diversity on small spatial scales. Abundances of microbial taxa varied substantially within the ridge, and highest values, though slightly lower than abundances found in level ice, were measured in void roof and floor ice, habitats that are unique to ridges. To the best of our knowledge, our current analyses are the

**Table 1 | Environmental and biological characterizations of previously sampled ridge habitats**

| Area | Year | Method | Focus | Ridge habitat | Max Chl-a | Dominant algal species found in/at ridges | Higher biomass associated with ridges | Study |
|---|---|---|---|---|---|---|---|---|
| Barents Sea | 1984–1988 | Divers | Flora | Surfaces of ice blocks in ridge flanks | ND | *Nitzschia frigida* & *Shionodiscus bioculatus* | ND | Syvertsen[8] |
| Barents Sea | 1986–1988 | Divers | Flora | Surfaces of ice blocks in ridge flanks | ND | *Nitzschia frigida* & *Shionodiscus bioculatus* & *Actinocyclus* sp. | Yes | Hegseth[16] |
| Baltic Sea | 2004–2005 | Divers | Flora | Ice blocks and void water | 40–50 mg m$^{-3}$ | *Thalassiosira baltica* & *Entomoneis paludosa* | Yes | Kuparinen et al.[18] |
| Fram Strait and Barents Sea | 2000–2005 | Divers | Fauna | Outer surfaces of ridge flanks | ND | ND | Yes | Hop and Pavlova[17] |
| Chukchi Sea and Canada Basin | 2005 | Divers | BGC & Fauna | Outer surfaces of ridge flanks (Ridge bottom ice) | ND | ND | Yes | Gradinger et al.[13] |
| Nansen Basin | 2015 | Divers and coring | BGC & Flora | Surface of ice blocks and core sections | 29.4 mg m$^{-3}$ | *Nitzschia frigida*, *Navicula* spp. & *Shionodiscus bioculatus* | Yes (besides a snow infiltration layer) | Fernández-Méndez et al.[15] |
| Barents Sea | 1986–2012 | Divers | Flora | Surfaces of ice blocks in ridge flanks | 39 mg m$^{-2}$ | *Actinocyclus curvatulus* & *Shionodiscus bioculatus* | Yes | Hegseth and Quillfeldt[129] |

first to investigate habitats from the interior of ridge keels, from winter to summer, and additionally demonstrate the accumulation of algae biomass inside ridges. We measured the highest Chl-a concentration (60.1 mg m$^{-3}$) in an upwards-facing ice block of a water-filled void (similar to image d in Supplementary Fig. 7), eight times higher than in level ice, confirming concentrations measured in previous ridge observations, which were limited to the outer flanks of the keels (Table 1). The Chl-a concentration of 5.3 mg m$^{-3}$ in the upper water-filled void on 3 July was more than twice as high as in the under-ice water at the same time and comparable to the only other recorded void water sample from a ridge much further south of our study site (low-salinity Bothnian Bay, Baltic Sea[18]).

We could not collect material from the surfaces of the outermost keel ice blocks (ridge flanks), but video footage from other parts of the ice floe (ledged/deformed ice) confirmed high algal biomass accumulations on upward-facing ice blocks (Supplementary Fig. 8 and ref. 37). The high Chl-a concentrations observed in the keel interior thus indicate an accumulation of algal biomass in ridges, which is typically overlooked since ridges are seldomly sampled. Currents, turbulence, nutrient advection, salinity differences, and different light conditions could cause differences in algal biomass between different parts of the ridge keels, with the interior of the ridge being least exposed to currents that affect keel flanks. A conceptual modeling study[38] and supported by recent field observations[39] suggest that ridges can be a conduit for light and thus affect light availability for primary producers. Fernández-Méndez et al.[15] found higher biomass on the ridge flanks located on the lee side of the keel, while Syvertsen[8] pointed out a stronger coloration on the current-facing side, perhaps related to the incorporation of algae or terrigenous particles, highlighting the complexity of ridge structures and their still underexplored effect on biology. The only exterior ridge samples we collected were ridge bottom ice, where Chl-a concentrations were low (0.4–1.8 mg m$^{-3}$) compared to the surrounding level ice (both SYI and FYI; 5.3–7.9 mg m$^{-3}$), likely resulting from the fast melt rate of the keel, especially in July when melt rates were highest[40].

Chl-a concentrations in water-filled voids were much lower in winter (0.02 mg m$^{-3}$) and spring (0.08 ± 0.05 mg m$^{-3}$) than in summer (2.6 ± 2.0 mg m$^{-3}$), but were twice as high as in upper surface water (winter: 0.01 ± 0.005 mg m$^{-3}$; spring: 0.04 ± 0.01 mg m$^{-3}$). Additionally, concentrations of Chl-a in winter and spring in ice blocks surrounding the voids were similar to the highest concentrations found in the interior of the level ice, likely representing incorporation of algal biomass when the ice formed. This indicates that ridges, and ridge voids in particular, given their sheltered location inside ridge keels, can form a refuge for algae to survive the polar night and seed spring growth, as previously suggested for multi-year ice[41]. In summer, interior ridge habitats are thermally and optically more protected compared to thinner level ice, reducing physiological stress on ice algae[15]. These protected locations also minimize downward transport and limit accessibility for grazers, reducing important algal loss mechanisms. The higher algal biomass in ridges compared to level ice is thus probably a result of the reduced loss and increased accumulation of algae in ridge voids and surrounding ice surfaces, as also suggested for ridge flanks (Table 1). Our biodiversity analyses that show similar alpha diversity indices across the different ice environments, i.e. similar within-habitat richness and evenness, further support this interpretation. However, the clear differences in beta diversity (based on 18S rRNA gene community composition analyses) highlight that ridges host distinct protist communities and indicate strong environmental filtering (Fig. 6). Further, gamma diversity was highest for ridges compared to FYI and SYI (Supplementary Fig. 5c), suggesting that the unique habitats provided by ridges select for certain genera (Supplementary Table 2). Hence, the distinct community structure found in the different ridge habitats is an important contributor to overall Arctic sea ice biodiversity.

The relative volume fraction and the large habitable ice surface area of ridges (see "Methods"), combined with the high Chl-a accumulation, explain why ridges can contain about 80% of the entire ice algal biomass of Arctic sea ice. This estimate is based on July measurements only and is hence only a snapshot from that season. During the spring ice algal bloom, the biomass in the bottom ice of both level ice and within the ridges was most

**Table 2 | Volume fractions, areal coverage, chlorophyll a (Chl-a), and particulate organic carbon (POC) concentration of different ice types in July**

| Ice type | Unit volume per ice type (m³ m⁻²) | Fraction of areal coverage | Fraction of total ice volume | Number of Chl-a samples | Chl-a (mg m⁻³) | Chl-a (mg m⁻²) | Number of POC samples | POC (mg m⁻³) | POC (mg m⁻²) |
|---|---|---|---|---|---|---|---|---|---|
| Ridge interior ice | 2.67 | 0.22 | 0.31 | 23 | 2.6 ± 0.9 | 1.5 ± 0.5 | 21 | 420 ± 60 | 250 ± 40 |
| Ridge exterior ice | 0.94 | 0.22 | 0.11 | 6 | 19 ± 9 | 4 ± 2 | 6 | 410 ± 100 | 80 ± 20 |
| Ridge voids | 0.29 | 0.22 | 0.03 | 3 | 3 ± 1 | 0.17 ± 0.07 | 3 | 100 ± 30 | 7 ± 2 |
| Ridge bottom ice | 0.1 | 0.22 | 0.01 | 5 | 1.2 ± 0.2 | 0.03 ± 0.01 | 5 | 390 ± 60 | 9 ± 1 |
| FYI interior ice | 1.2 | 0.5 | 0.31 | 60 | 0.9 ± 0.1 | 0.57 ± 0.09 | 54 | 280 ± 30 | 170 ± 20 |
| FYI bottom ice | 0.1 | 0.5 | 0.03 | 5 | 3.9 ± 0.9 | 0.20 ± 0.04 | 5 | 950 ± 220 | 50 ± 10 |
| SYI interior ice | 1.3 | 0.28 | 0.19 | 68 | 0.8 ± 0.1 | 0.30 ± 0.04 | 73 | 760 ± 70 | 270 ± 30 |
| SYI bottom ice | 0.1 | 0.28 | 0.01 | 4 | 3.1 ± 0.7 | 0.09 ± 0.02 | 4 | 470 ± 100 | 13 ± 3 |

Areal coverage of ridges, first- (FYI) and second-year ice (SYI) are based on laser scans of a 40 km² area around the MOSAiC ice floe. Concentrations are given as mean ± standard error (SE).

certainly higher than what we measured in this study, but their relative contribution to Chl-a per sea ice area may, however, still be comparable, due to the larger volume of habitable space in ridges. It is important to note that the ridges we studied were formed prior to the ice algal spring bloom. Therefore, the high Chl-a biomass suggests in situ growth rather than biomass introduced with the incorporation of level ice as the main contributor to biomass build-up in ridges. However, biomass from the previous growing season locked inside the level ice that formed the ridge likely contributed an important algal seed stock for the developing ridge communities. The calculated POC standing stocks in the different ice types also show that the relative amount of POC per ice type areal coverage was higher in the ridge than in FYI and SYI. Surprisingly, the concentration of POC in SYI was much higher in the interior than in the bottom layer (Table 2), suggesting that a substantial fraction of the carbon in SYI is likely stored material. This observation is supported by the C:Chl-a ratio, which was highest in SYI (432), compared to much lower values in FYI (148) and the ridge (31). These ratios indicate that ridges provide more habitable space supporting active algal growth, while level ice may contain relatively more stored organic material from previous growing seasons, and depending on the origin and age of the ice, also material incorporated where the ice formed, such as sediment, flora and fauna from the Siberian Shelf (Supplementary Fig. 9[42]). In addition, non-phototrophic organisms (such as heterotrophic nanoflagellates and bacteria) also contribute to the POC pool, further corroborating our interpretation that Chl-a may be a better indicator of actively growing in situ algal biomass in sea ice compared to POC.

**Biotic response to freezing of water-filled voids**

The seasonal interior freezing and consolidation of the ridges were substantial, exemplified by a reduction in macroporosity from 22% ± 8% to 2% ± 4% in the ridge keel from January to July, and unconsolidated areas of the ridge were restricted to the flanks[27]. Freezing of voids was likely caused by infiltration of low-salinity meltwater[26], possibly linked to a large melt pond draining event at the beginning of July[31]. This meltwater-driven freezing of the water-filled voids inside the ridge caused a strong biological response that differed between eukaryotic and prokaryotic community members. Whereas Chl-a concentrations were lower (Fig. 3a), both bacterial abundances, in particular HNA-bacteria, and bacterial production were higher in ridge ice samples when voids were frozen compared to water-filled (Mann–Whitney $U$ test; $p = 0.02$; Fig. 3f and Supplementary Fig. 6). This finding, together with strong shifts in metabolic pathways, suggests a change from a predominantly autotrophic to a predominantly heterotrophic community with the freezing of the water-filled voids. The increase in HNA-bacteria was accompanied by a higher relative abundance of Gammaproteobacteria, known to include fast-growing species. From the 16S rRNA gene amplicon dataset, we found that bacterial taxa known for their carbon degradation potential, such as Gammaproteobacteria of the genera *Colwellia*, *Paraglaciecola*, and *Oleispira* dominated the frozen voids with *Colwellia* being most prominent. Accordingly, we observed that a high proportion of the MAGs belong to the class of Gammaproteobacteria in ridge ice samples when the voids were frozen (Fig. 8a). The higher relative proportion of carbon-processing genes associated with Gammaproteobacteria in frozen voids (31%) compared to 18% in the ice surrounding water-filled voids indicates that freezing does not suppress and may even select for taxa involved in carbon degradation (Supplementary Table 6). The higher prevalence of genes encoding enzymes for the breakdown of substrates of varying complexity when voids are frozen (Fig. 8b) suggests that bacterial communities are metabolically versatile, capable of utilizing a broad range of carbon sources found in the ridge ice[43,44]. Together, these changes in the metabolic potential imply that ridges, particularly when voids are frozen, provide conditions for increased carbon turnover.

Further detailed analysis revealed that three *Colwellia* eco-types (with two ASVs each) were abundant in different samples, with a relative abundance of up to 62%. Such high dominance is only known from environments with specific substrate/nutrient availability, such as deep-sea hydrothermal vents, (SUP05-clade (60%)[45]) and permafrost soil,

(*Bacteroidia* (54%)[46]), specific host-microbe interactions (*Vibrio fischeri*—bobtail squid symbiont (100%)[47]) or associated with phytoplankton bloom conditions (e.g., Barents Sea ice edge bloom, *Polaribacter* (72%)[48]). These are all examples that highlight how fast-growing substrate-specific bacteria can dominate prokaryotic communities when conditions allow, often as a result of the excess of a specific food/energy substrate. *Colwellia* species have been found in high abundances on particulate organic matter in the central Arctic as well as in response to particulate organic matter addition[49,50]. The functional annotation of a MAG (ASM4763325v1) assigned to *Colwellia* from a frozen void sample supports this role and suggests a heterotrophic, facultatively anaerobic metabolism with a large potential to degrade algal-derived organic matter. We identified several genes that encode different carbohydrate-active enzymes (including cellulases, β-glucosidases, xylanases, and amylases) and multiple secreted hydrolases, consistent with extracellular polysaccharide depolymerization. Measurable rates for some of these polysaccharide hydrolases in the central Arctic[49], particularly in waters with high proportions of *Colwellia*, indicate a key biogeochemical importance in carbon degradation of these taxa. Notably, the *Colwellia* ecotypes observed in ridge ice samples were co-located with specific 18S rRNA ASVs with significant correlation between the *Colwellia* ecotypes and diatom sequences (*Colwellia*-1 and *Navicula* [$p < 0.002$], *Colwellia*-2 and *Shionodiscus* [$p < 0.0001$], *Colwellia*-3 and *Attheya* [$p < 0.0001$]). This specificity suggests ecotype-level niche differentiation within *Colwellia*, potentially driven by the availability and composition of algal-derived organic substrates as previously shown in studies on metapangenome, comparative genomics, and phenotype arrays of *Colwellia*[51,52]. Consistent with this, the 12 MAGs taxonomically assigned to *Colwellia* showed a large variability in their repertoire of carbohydrate-active enzymes, ranging from highly specialized to broad-range degraders (Supplementary Table 7). We identified these specific bacteria-algae interactions both when voids were water-filled and when they were frozen, implying a tight coupling between the carbon-processing capabilities in the *Colwellia* ecotypes and the production and turnover of the diatoms that is sustained under the changed conditions when the voids were frozen.

The meltwater-driven freezing of water-filled voids exposes the ridge biota to two challenges: a strong shift in salinity and a shift from a liquid phase to a solid phase. Even though diatoms have a mechanism to tolerate and adapt to osmotic stress[53], higher mortalities have been shown for *Navicula* and *Nitzschia* during the ice melt period, when changes occur rapidly[54,55]. We identified a strong shift in the community composition of protists upon freezing, with decreases of mainly diatoms and dinoflagellates (Fig. 3c, d). This decrease indicates that these organisms were either flushed away with the inflow of meltwater, mixed and diluted during the freezing process, or degraded by heterotrophic bacteria that increased in abundance upon freezing. Void freezing also makes ridges less suitable for larger animals like amphipods and juvenile polar cod that seek ridge voids as feeding and hunting grounds and hiding spaces (Supplementary Figs. 2 and 7[17,56]). Sea ice meiofauna and polar cod can tolerate a wide range of salinities[56,57], but the energetic costs of osmoregulation associated with low-salinity conditions are avoided through downward migration along the ridge keel, as shown for various ice meiofauna and ice-associated amphipods. Additionally, some organisms were potentially transported into the ridge with surface meltwater, as indicated by the increased abundance of Betaproteobacteria (*Herbaspirillum*), often associated with terrestrial environments, and evidence for increased sediment material of terrestrial origin in frozen void samples. Higher concentrations of lithogenic silica (Supplementary Fig. 10) and enhanced particulate iron and magnesium (Supplementary Fig. 11) indicate increased sediment material after freezing. This finding suggests that terrestrial particles that were incorporated into the MOSAiC ice floe when it formed over the Siberian coast in fall 2019[42] were transported with meltwater into the ridge interior. This incorporated material is released again upon continued ridge melt on its way to the marginal ice zone, potentially contributing nutrients (e.g., silicic acid) to sustain the large phytoplankton (e.g., diatom) blooms along the ice edge[58–60].

Our study reveals high habitat diversity and unique microbial communities within sea-ice ridges, that are potential reservoirs for ice-associated organisms in winter and support high biomass accumulation in summer (Fig. 9). As communities differed strongly over small spatial and temporal

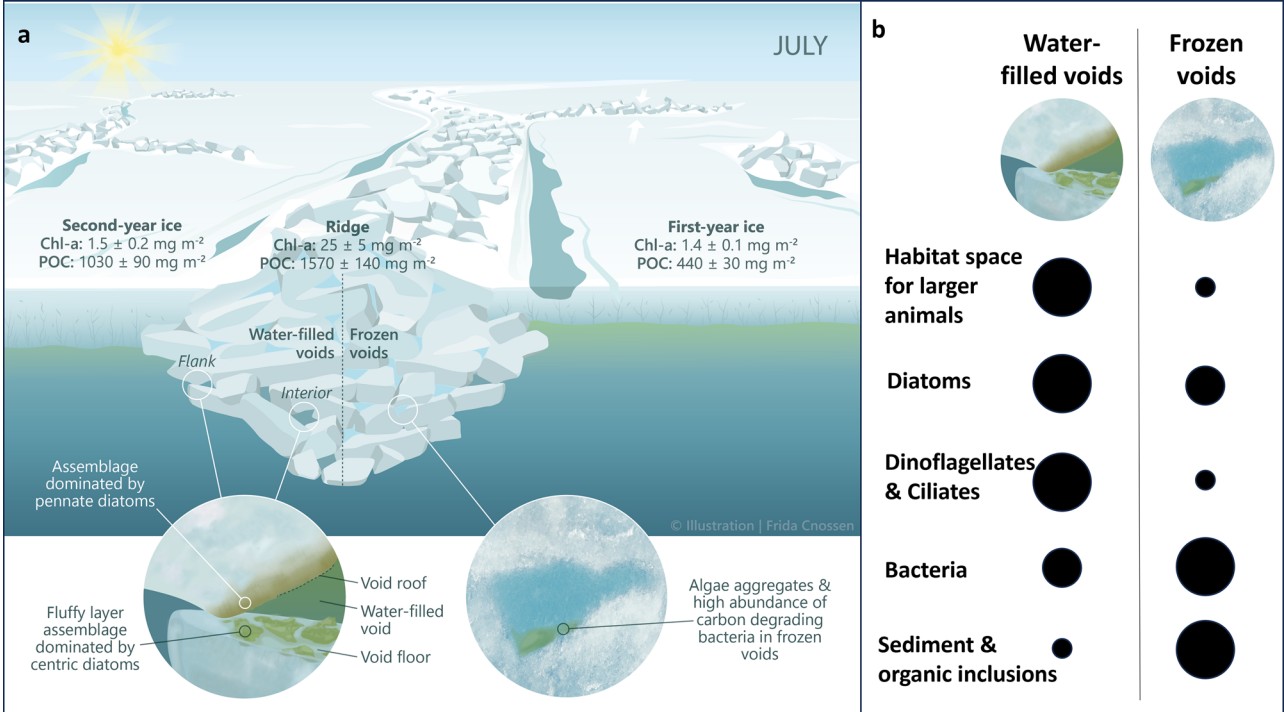

**Fig. 9 | Chlorophyll *a* and particulate organic carbon in different ice types.**
**a** Average integrated chlorophyll *a* (Chl-a) and particulate organic carbon (POC) values (per m² sea ice) of the different ice types and ecosystem changes as a response to freezing of the water-filled voids that occurred in July 2020 during the MOSAiC expedition. **b** Black circles indicate the relative magnitude of the various variables before and after the freezing. Credit: Frida Cnossen.

**Table 3 | Sampling events and depth for biological and physical variables at the three ridges investigated during the MOSAiC expedition**

| Ridge | R1 | | R2 | | R3 | | | | |
|---|---|---|---|---|---|---|---|---|---|
| Date (2020) | 10.01. | 24.01. | 22.04. | 05.05. | 03.07. | 10.07. | 18.07. | 24.07. | 24.07. |
| Specifics | | | | | | Frozen voids | | | Ridge flank |
| A—ice void roof | X (153–184) | X (157–204) | X (122–132) | X (118–128) | X (179–189) | **X** (74–84) | | | X (172–184) |
| B—upper void | X (184–214) | X (204–224) | X (132–147) | X (128–145) | X (189–199) | **X** (95–105) | | | X (184–196) |
| C—ice void floor | X (214–243) | X (224–261) | X (147–157) | X (145–155) | X (199–209) | **X** (121–131) | | | X (196–207) |
| D—ice void roof | X (349–385) | | | | X (277–287) | | **X** (170–180) | **X** (128–139) | |
| E—lower void | | | | | X (287–369) | | **X** (192–202) | **X** (193–203) | |
| F—ice void floor | X (420–460) | | | | X (366–376) | | **X** (284–294) | **X** (267–279) | |
| G—algae inclusion | | | | | | **X** (383–393) | **X** (183–188) | | |
| H—ice ridge bottom | | | | | X (550–560) | X (391–401) | X (511–522) | X (503–513) | X (425–435) |
| UiW-ridge | | | | | | X | X | X | |
| UiW-FYI | | | | | | X | X | X | |

The letters specifying the type of ridge sample correspond to those provided in Figs. 3 and 10. The X indicates when these samples were taken. A bold **X** indicates samples that were taken when the water-filled voids were frozen, and hence, ice sections corresponding to the frozen void and the roof or the floor of the void were not as clearly defined as for the other coring events when water-filled voids were encountered. For R1, the sampled ice sections were between 30 and 50 cm and for R2 and R3 sections were 10 cm long. Sampling depths (cm) from the water surface (corrected for ice draft) are given in brackets.

*UiW* under-ice water.

scales, future research should use approaches that can contrast the outer, unconsolidated parts and the inner, seasonally and event-driven consolidated parts of ridges. Increased bacterial activity in the consolidated ridge ice in summer indicates that freezing may enhance rather than inhibit prokaryotic carbon degradation, consistent with the observed shifts in metabolic potential linked to carbon processing. The contribution of ridges to the Arctic marine carbon cycle remains unresolved but could be substantial due to their large habitat volume and high biomass reservoir in terms of Chl-a. Our estimates show, for example, that ridges contain up to 80% of the ice algal biomass, thus being important for central Arctic primary production, but also as a food source for higher trophic levels. Given this potentially large contribution to overall Arctic sea-ice biomass, it is crucial to focus more sea-ice research on ridges to understand how the Arctic ecosystem functions and predicting how it will change in the future. Here, research should also look beyond the unicellular components to elucidate currently unknown food web interactions and related biogeochemical processes within the various ridge habitats.

## Methods
### Study area
The year-long MOSAiC ice drift (October 2019 to September 2020), with the research vessel RV Polarstern serving as the base, started in the eastern Eurasian Basin and crossed the Amundsen and Nansen basins towards the Fram Strait (Fig. 1) (e.g., refs. 61,62). One dedicated research project (Ridges—safe HAVens for ice-associated flora and fauna in a seasonally ice-covered Arctic Ocean (HAVOC)[63]) performed detailed and interdisciplinary observations of ridges during MOSAiC. During the drift, three different ridges were sampled at different times of the year. The changes between ridges were necessary, as logistical challenges and ice dynamics prevented the sampling of the same ridge throughout the entire period (see below), highlighting the difficulties associated with studying ridges. The first ridge (R1) was investigated in winter, the second ridge (R2) was investigated in spring, and the third ridge (R3) was investigated in summer (Figs. 1 and 9 and Table 3). Based on their macrostructural physical properties, the three ridges were similar in characteristics. They formed during the MOSAiC drift (i.e. they were of similar age) and were composed of thin ice blocks, with similar sail heights (1–2 m) and average keel depths (3.2–4.3 m).

The ridge in winter (R1, named Fort Ridge in the field) was approximately 90–100 m long and 20–30 m wide. It was estimated to have formed in September–October 2019 based on surface laser scanning[64]. The maximum keel depth measured by drilling was 6.8 m (7.4 m from sonar), the maximum sail height estimated from drilling was 1.3 m (closer to 3 m from surface laser scanning), and the average snow depth was 0.4 m. The rubble macroporosity was 26–33% determined from drilling in January, and 9–46% derived from three ice mass balance buoys, installed in the middle and flanks of the ridge, from January to March, showing high spatial variability in macroporosity. R1 was sampled twice for biological variables on January 10 and 24 at around 87°N in the eastern Amundsen Basin (Table 3 and Fig. 1). Because R1 was not logistically accessible in spring and summer, other ridges were chosen for sampling. The ridge studied in spring (R2, named Ridgey McRidge Face in the field) formed in mid-March during a period of repeated dynamic ice movements[29]. No instruments were installed in R2, but void water and adjacent ice (void roof and floor) were sampled twice for physical and biological variables in late April and early May at about 84°N (Fig. 1 and Table 3). From surface laser mapping, the typical sail height was 1.2 m, with a maximum of 2.2 m. At the location of sampling, the snow depth ranged widely between 0.12–0.8 m in late April and 0.3–0.44 m in May, but was thick compared to level ice with an average snow depth of 0.14 m[65]. R2 disintegrated during a very dynamic period around mid-May and could not be investigated further. The ridge sampled in summer (R3, named Jaridge in the field) formed in early February and was studied repeatedly over a period of about 1 month (late June to late July 2020) when the ice floe drifted from 82 to 80°N (Fig. 1 and Table 3)[26,40]. Based on extensive sonar mapping, the ridge had an average keel bottom depth of 4.6 m and a keel width of 42 m, with a mean sail height of 0.5 m and a sail width of 15 m estimated from drilling. The average snow depth above the ridge in early July was 0.5 m and the average macroporosity of the keel and the rubble was 4% and 15%, respectively. Based on drillings, this ridge had lower macroporosity than R1 in January and had experienced substantial consolidation. There was also substantial melt of the ridge keel, with 1.7 m maximum melt loss at the keel and 4.5 m at the ridge flanks[40]. The ridge melted about 3 times faster than the surrounding level FYI in July, mainly due to enhanced bottom melt. Sampling for biological variables of void water, void roof and floor ice, ridge bottom ice and under-ice water was done at four time points in July at different locations of the ridge (Table 3

and Fig. 10). The entire MOSAiC ice floe, including R3, disintegrated on 31 July, less than 50 km from the ice edge[66], and no further sampling was possible.

## Sampling of ice and water from ridges and level ice

Ice cores for temperature and salinity measurements as well as biogeochemical variables were extracted with a 9-cm (Mark II) internal diameter ice corer (Kovacs Enterprise, USA). Ice temperature was measured in situ using a Testo 720 thermometer in drill holes with a length of half-core diameter at 5–10 cm vertical resolution. Ice bulk salinity was measured from melted ice core sections using a YSI 30 conductivity meter (the conductivity is converted to salinity and reported on the Practical Salinity Scale 1978, PSS-78, which is dimensionless). The relative brine volume fraction of each section was calculated following Cox and Weeks[67] and Leppäranta and Manninen[68] for in situ conditions using the ice temperature profile measured in the field and the bulk salinities. We also used temperature measurements from ice mass balance buoys (SIMBA[69], DTC[70]) to study the temporal evolution of thickness and temperature. These buoys had a chain of temperature sensors with a 2–4 cm sensor spacing and recorded temperature every 6 h with an accuracy of 0.1 °C. Three ice mass balance buoys were installed at R1, and one ice mass balance buoy was installed at R3.

Ice cores collected for biogeochemical variables were cut into 10 cm long sections in the field and collected in sterile plastic bags, with the focus on the three habitats: the ice of the roof and the floor of water-filled voids, the bottom of the ridge, and, when present, the frozen void and algae inclusions. Biogeochemical variables were, when possible, derived from pooled ice core sections of three replicate cores (R3), and during challenging weather periods (R1 and R2) from single ice cores. The core sections were kept dark and cool, transferred to the laboratory on board and melted in the dark after the addition of filtered seawater: 50 mL of 0.22 μm filtered seawater was added per cm of sea ice thickness, and the sea ice samples melted within 24–36 h in the dark at around 4 °C. When possible, the water (20–30 L) inside the voids, below the ridge and below level ice, was sampled using a manual bilge pump with a silicon tube with a diameter of 20 mm into prewashed polyethylene containers. When two different void locations were sampled during the same sampling event (R3: 03.07.2020), due to the complexity and interconnectivity of voids, we cannot exclude that the sampled water was only from one individual void and potentially a mix from neighboring, connected voids. However, due to the large vertical distance (1 m) between the upper and lower voids, we can exclude vertical cross-contamination. From both melted sea ice and water samples, sub-samples were taken for determination of inorganic nutrients, biogenic silica (BSi), particulate organic carbon (POC), chlorophyll *a* (Chl-a), bacterial production (BP) and abundance and diversity estimates of protists and bacteria through flow cytometry (FCM), light microscopy and molecular analysis, as described in more detail below.

Comparative data from first-year and second-year level ice (FYI and SYI) used in this study are from ice cores taken as part of the MOSAiC main level ice coring program, where ice cores were retrieved, sectioned, and treated in a similar way as described above. For further details, see Nicolaus et al.[61] and Fong et al.[62].

## Nutrient analysis

Samples collected for nutrient measurements were analyzed onboard (January–May 2020), and samples stored frozen (June–July 2020) were analyzed at a shore-based lab. In either case, analyses were conducted colorimetrically using an AA3 continuous flow auto analyzer (Seal Analytical) following best practices adopted from GO-SHIP recommendations[71,72].

## Biogenic and lithogenic silica

Bio- (BSi) and lithogenic (LSi) silica were analyzed using a combination of previously published methods. Given the high lithogenic silica content of ice and water samples during this campaign, a commonly used sequential sodium hydroxide and hydrofluoric acid digestions for water-column BSi and LSi would have been unsatisfactory (e.g., ref. 73). Sodium hydroxide digestions can dissolve some LSi in field samples; many studies assume this

to be up to 15% and apply a correction factor (e.g., ref. 73). Given the expected (and realized) LSi loads, a 15% LSi correction would overcorrect the BSi signal. Hence, we applied a more laborious time-course digestion protocol, which uses 0.1 molar sodium carbonate (as done previously for turbid Arctic coastal water samples in Varela et al.[74]) and allows for better isolation of the BSi signal from the solubilized LSi. This is similar to time-course digestions used in marine sediments[75]. Post sodium carbonate digestion, the LSi in the remaining material was solubilized using dilute 2 molar hydrofluoric acid and quantified as dissolved silicic acid (colorimetric molybdate method). BSi is determined as the y-intercept of the time-course[75] and the total LSi reported is the sum of phases quantified in both the sodium carbonate- (total liberated Si minus that from only BSi) and hydrofluoric acid digestions.

## Particulate organic carbon (POC)

Between 0.3 and 2 L of the sampled water or melted ice was filtered onto precombusted Whatman GF/F filters under a low vacuum pressure (≤30 KPa). Filtered quantity was adjusted depending on the coloration of the filter. The filters were folded and frozen at −80 °C until analysis. Before the analysis, filters were first dried overnight at 60 °C, and afterwards acid-fumed with hydrochloric acid for 24 h before they were dried again for 24 h at 60 °C. For the measurement, the dried filters were then packed in tin capsules and measured with an elemental analyser (Flash 2000, Thermo Scientific, Bremen, Germany) at the stable isotope facility of the joint research unit Littoral, Environment and Societies (CNRS-University of La Rochelle), France.

## Chlorophyll *a* (Chl-a) analysis

Water samples and melted sea ice samples were filtered on 25 mm GF/F filters (Whatman) using a low-pressure vacuum. When possible, triplicates were taken (water samples), while one sample was taken per ice core section, and volumes ranged between 0.2 and 0.5 L for ice samples and 0.5 and 2 L for water samples. The filters were then transferred into 2 mL Eppendorf tubes and frozen at −80 °C. Due to a mistake, the filters from ridge samples were dried overnight at 60 °C for transport and, following arrival at the lab at the University of Bergen, kept frozen at −80 °C. As this procedure compromised the samples, a thorough test was performed to evaluate the potential effect of the drying on Chl-a concentration, which is described in detail in Supplementary Fig. 9. In short, independent of algae biomass, the dried filters had consistently 1.8 times less Chl-a than the non-dried standard treated filters, and hence this correction factor was applied to the ridge samples (Supplementary Fig. 12). For further analysis, the samples were extracted in 90% acetone overnight at 4 °C and subsequently analyzed on a calibrated Turner Design 10-AU fluorometer (Turner Designs, USA), including an acidification step (1 M HCl) to determine phaeopigments[76]. All filtration, extraction, and measurement steps were performed with dimmed lights.

## Flow cytometry counts

Triplicate subsamples of 1.8 mL from water or melted sea ice samples were fixed with 36 μL 25% glutaraldehyde (0.5% final concentration) for 2 h at 4 °C, flash-frozen in liquid nitrogen, and stored at −80 °C until analysis. Abundances of pico- and nanosized phytoplankton, and heterotrophic nanoflagellates (HNF) were measured using an Attune® Acoustic Focusing Flow cytometer (Applied Biosystems by Thermo Fisher Scientific, Waltham, MA, USA) with a syringe-based fluidic system and a 20-mW 488-nm laser. Autotrophic microorganisms were counted directly after thawing of the sample, and the different phytoplankton groups were discriminated based on their red fluorescence (BL3) vs. orange fluorescence (BL2) and orange fluorescence (BL2) vs. side scatter (SSC) as in Paulsen et al.[77]. For the measurement of HNF abundance, once thawed, the samples were stained with a green-fluorescent nucleic acid dye (SYBR Green I; Molecular Probes, Eugene, OR, USA) for 2 h in the dark, and then 1.5 mL measured at a flow rate of 500 μL min$^{-1}$ following the protocol of Zubkov et al.[78]. Bacteria were measured on a FACS Calibur (Becton Dickinson) flow cytometer. Samples were first thawed, diluted 10 times with 0.2-μm-filtered TE buffer (Tris

10 mM and EDTA 1 mM, pH 8), stained with SYBR Green I, and incubated for 10 min at 80 °C in a water bath[79]. The samples were counted at a flow rate of around 60 μL min$^{-1}$ and different groups discriminated on a biparametric plot of green fluorescence (BL1) vs. side scatter (SSC) to distinguish different bacterial groups, including low nuclear acid (LNA) and high nuclear acid (HNA) bacteria. A figure exemplifying the gating strategy is provided in the Supplementary Information (Supplementary Fig. 13).

### Light microscopy
For the preservation of protist samples, 200 mL of water and melted sea ice were fixed with a Lugol-formaldehyde mixture with a few drops (2–3 mL) of acidic Lugol solution and hexamethylenetetramine-buffered formalin solution at a final concentration of 1% and kept at 4 °C in the dark. Before counting, the cells were settled for 48 h in Utermöhl sedimentation chambers (HYDRO-BIOS©, Kiel, Germany) and identified and counted using an inverted light microscope.

### Bacterial production
Bacterial production was measured based on the incorporation of 3H-leucine according to Smith and Azam[80]. Samples (seawater or melted sea ice) were distributed into four replicates of 1.5 mL using 2 mL Eppendorf vials. A total of 80 μL of 100% trichloroacetic acid (TCA) was immediately added to one replicate, which served as a control. All replicates were then incubated with 25 nM 3H-leucine (final concentrations) for 2 h at in situ temperature, before they were stopped by adding 80 μL of 100% TCA. Samples were kept cold at 4 °C until analysis. Samples were centrifuged for 10 min at 14,800 rpm, the supernatant was removed, and the pellet was subsequently washed with 1.5 mL 5% TCA. This step was repeated twice, and after the last centrifugation step, once the supernatant was removed, 1.5 mL of scintillation liquid (Ultima Gold) was added. The radioactivity in the samples was then counted on a Perkin Elmer Liquid Scintillation Analyzer Tri-Carb 2800TR. The measured leucine incorporation was converted to μg carbon incorporated per L per day (presented as mg C m$^{-3}$ d$^{-1}$), using a conversion factor of 1.5 kg C per mole of incorporated leucine, based on the specific activity of the isotope and the constants 1797 (grams of protein produced per mole of incorporated leucine) and 0.86 (the weight ratio (g:g) of total C:protein in bacteria) according to Simon and Azam[81], assuming no isotope dilution[82].

### DNA extraction, PCR, Illumina 16S/18S rRNA gene and metagenome sequencing
Both melted sea ice and water samples were filtered through a Sterivex 0.22 μm filter or, when volumes were <500 mL (three sea ice samples), onto 0.22 μm Durapore filters. The filters were directly flash-frozen in liquid nitrogen, stored at −80 °C, and at the end of the campaign, shipped to the University of Bergen on dry ice. DNA from Sterivex filters was extracted using the Qiagen PowerWater DNA kit, and the DNA from Durapore filters was extracted using the QIAGEN DNeasy Power Soil kit following the manufacturer's instructions. The extracted DNA was stored at −20 °C until PCR amplification for amplicon sequencing (16S and 18S rRNA gene) or shipment to the Joint Genome Institute (CA, USA) for metagenome sequencing using 151 base pair paired-end reads with an Illumina NovaSeq S4 device, retrieving 42 HAVOC metagenomes. Metagenome data were assembled using the JGI MAP pipeline[83], where samples were filtered for quality with BBDuk, error corrected using BBCMS, assembled using SPAdes[84], and reads mapped back to contigs using BBMap (v38.86[85]). Metagenomic data were analyzed to compare functional gene profiles across samples. Annotated gene abundances were grouped according to Clusters of Orthologous Groups (COG) pathways. Gene abundance tables were normalized to account for differences in sequencing depth. Principal Component Analysis (PCA) was used to explore patterns of variation in COG pathway abundance among samples. Metagenome assembled genomes (MAGs) recovered from the HAVOC metagenomes (PRJNA1160706), as described in ref. 86, were analyzed using METABOLIC v4.0[87] to infer metabolic potential and using dbCAN4[88] to identify and annotate

carbohydrate-active enzymes using the Carbohydrate-Active EnZymes database (CAZy)[89].

For amplicon sequencing a two-step nested PCR approach using HotStarTaq Master Mix (Qiagen) was applied targeting the bacterial/archaeal 16S rRNA gene (519 F: CAGCMGCCGCGGTAA[90]; and 806RB: GGACTACNVGGGTWTCTAAT[91]) and targeting the eukaryotic 18S rRNA gene (528iF: GCGGTAATTCCAGCTCCAA[92]; and 964iR: ACTTTCGTTCTTGATYRR[92]). The first PCR, performed in triplicates, was done at the following conditions: initial denaturation of 15 min at 95 °C, followed by 25 cycles of 95 °C for 20 s, 55 °C for 30 s, and 72 °C for 45 s and a final extension step of 72 °C for 7 min. With the second PCR, unique eight-nucleotide barcodes were added to the first PCR products using the following conditions: initial denaturation of 15 min at 95 °C, followed by 12 cycles of 95 °C for 20 s, 62 °C for 30 s, and 72 °C for 30 s, followed by a final extension step of 72 °C for 7 min. Finally, PCR products were cleaned with Agencourt AMPure XP magnetic beads (Beckman Coulter Inc., CA, USA), quantified using Qubit 3.0 Fluorometer, and pooled in equimolar concentrations before sending for sequencing to the Norwegian Sequencing Center (Oslo, Norway) using the MiSeq Reagent Kit v3 (Illumina, CA, USA). All ridge-associated sequences are available at the European Nucleotide Archive (ENA) under project number PRJEB90852. 18S sequences from seawater (28), first-year ice (FYI: 13), and second-year ice (SYI: 10) were obtained from the Alfred-Wegener Institute and can be found at ENA under the project number PRJNA895866.

In total, 34 ridge samples (including ice, void water, and seawater just below the ridge) were analyzed for 16S and 18S rRNA gene-based community composition, and in addition, 28 seawater samples, 13 FYI, and 10 SYI were analyzed for 18S community composition. The retrieved sequence data were processed using DADA2 (Divisive Amplicon Denoising Algorithm 2, version 1.32.0[93]) in R (version 4.4.1[94]). In short, sequences were trimmed, filtered based on quality scores, dereplicated, and amplicon sequencing variants (ASVs) inferred. Forward and reverse reads were denoised and merged before chimeric sequences were removed, and taxonomy was assigned using the 16S silva database (version 138[95]) and the 18S pr2 database (version 4.12.0[96]) for 16S and 18S sequences, respectively. The resulting ASV tables (16S and 18S), along with taxonomic annotations, were used for all subsequent diversity and community composition analyses. Relative abundance was calculated per sample by dividing each ASV count by the total number of reads.

### Statistical analysis
Statistical analyses were done using the R package vegan and the programs Primer-E version 6 (Plymouth, UK) and Canoco 5[97]. Among others, Bray-Curtis dissimilarities, principal component analysis (PCA), and linear regressions were calculated using these tools. The connections between environmental variables were determined using PCA, performed on normalized data using centered log-ratio transformation[98]. The coordinates of the scores and loadings for the PCA plots were illustrated using GraphPad Prism version 10.1.2 for Windows (GraphPad Software, USA, http://www.graphpad.com/). For the ordination analysis (NMDS) of the sequencing data, the relative abundance at the genus level was used, and the table was normalized using a square-root transformation before Bray–Curtis dissimilarity was calculated to assess community similarity among samples. For 16S datasets, all chloroplast sequences and for 18S datasets, all Crustacea sequences were removed prior to analysis. For the network analysis, Bray-Curtis dissimilarity was calculated using the R package vegan, converted into a similarity matrix (1 − distance), and an adjacency matrix was generated by applying a similarity threshold of 0.4. An undirected network graph was constructed with the R package igraph, and node degree (the number of sample connections) was calculated. The network was visualized using the R package ggraph to assess sample connectivity.

### Estimate of ice surface area and brine volume in ridges
To our knowledge, there have been no attempts so far to estimate the ice surface area in ridges (the sum of all ice block surfaces in a ridge keel).

Based on typical values of first-year ice ridge keel depth (6–8 m[4,7]), rubble macroporosity (30%[7]), block thickness (0.2–0.4 m[4]), and a block length to thickness ratio of 3.5, we estimate that the initial per unit surface area of ridge keels is about 20–30 times larger than for level ice. The ratio of the surface area of the ice blocks in the rubble $S_{rbl}$ and the horizontal projection of the ridge keel $S_{li}$ can be estimated as (Eq. (1)):

$$\frac{S_{rbl}}{S_{li}} = \frac{h_{rbl}(1-\mu)}{h_b l_b}(l_b + 2h_b) \tag{1}$$

where $h_{rbl}$ is the rubble thickness, $\mu$ is the rubble macroporosity, $h_b$ and $l_b$ are the block thickness and length. This will decrease with consolidation, but means that younger unconsolidated ridges with a 5% aerial fraction would typically provide an equal ice surface area as that of all of the level ice.

The brine volume of submerged ice blocks in the unconsolidated part (rubble) of the keel also adds to the habitable space in ridges, given that they are close to the seawater freezing point and are thus more porous than the colder level ice. The ratio of the total brine volume of the ridge $V_{b,rdg}$ and level ice $V_{b,li}$ can be estimated as (Eq. (2)):

$$\frac{V_{b,rdg}}{V_{b,li}} = \frac{h_{cl}}{h_{li}} + \frac{h_{rbl}(1-\mu)}{h_{li}}\frac{v_{b,rbl}}{v_{b,li}} \tag{2}$$

where $h_{cl}$, $h_{rbl}$ and $h_{li}$ are the thicknesses of the consolidated layer, the rubble, and level ice, respectively, and $v_{b,rbl}$ and $v_{b,li}$ are the brine volume fractions of rubble and level ice, respectively.

The excess brine volume fraction of ridges can be estimated as a function of ice temperature and salinity from, e.g., Frankenstein and Garner[99], giving rubble a brine volume of 10% and level ice a brine volume of 5% assuming an ice surface temperature of −10 °C, rubble salinity of 4, and first-year ice salinity of 5. For a typical level ice thickness of 1.5 m[100] and rubble porosity of 30%, Eq. (2) gives a ratio of ridge and level ice brine volume of around 8 during winter. After the onset of melt, both the ridges and the level ice become isothermal, and this leads to a lower ratio of unit brine volume of around 4 between ridge rubble and level ice.

On the large scale, observations indicate that the aerial coverage of ridged ice is around 30–40%[3,101], but up to 60% in heavily deformed areas[2]. Hence, the ice surface area in ridges can be multifold compared to that provided by level ice.

## Estimates of ridge and level ice volumes (FYI and SYI) to calculate Chl-a and POC concentrations in different ice types

For such estimates, it is practical to estimate unit volumes, i.e., volumes expressed in meters (m³/m², volume per unit area). These estimates can then be upscaled using areal fractions of ridge keels, first- (FYI) and second-year ice (SYI) from airborne surveys (here we used airborne laser scanner surveys). For most estimates here, we used an ice depth threshold of 2.5 m, with ice of a smaller depth not being considered a ridge. We also ignored the above-water contribution from ridges (sail) and level ice (freeboard) to the unit volume estimates, as they cannot serve as an algae habitat. Estimates are provided for the period from 25 June to 3 July 2020 from R3.

Key values include:
- Ridge block thickness $h_b = 0.3$ m (thickness tape measurements)
- Ridge rubble porosity $\mu_{rbl} = 17\%$ (drilling, 25 June–3 July, $n = 8$)
- Ridge consolidated layer thickness $h_c = 2.2$ m (drilling, 25 June–3 July, $n = 8$)
- Mean ridge keel depth $h_k = 3.9$ m (ROV multibeam measurements, the whole ridge)
- Mean level ice depth ($h_{fyi} = 1.4$ m and $h_{syi} = 1.5$ m, IMBs, $n = 7$)
- Areal coverage of ice type ($a_{rdg} = 22\%$, $a_{fyi} = 50\%$, $a_{syi} = 28\%$ from ALS[102]

Estimates for each ice type:

Ridge void unit volume (rubble thickness by rubble porosity):

$$v_{v,rdg} = (h_k - h_c)\mu_{rbl} = 0.29 \text{ m} \tag{3}$$

Ridge ice unit volume (consolidated layer and rubble ice fraction):

$$v_{i,rdg} = h_c + (h_k - h_c)(1 - \mu_{rbl}) = 4.1 \text{ m} \tag{4}$$

Ridge exterior (10 cm from surface) unit volume (block number by doubled layer thickness):

$$v_{ext,rdg} = (h_k - h_c)\frac{1 - \mu_{rbl}}{h_b}2 \cdot 0.1 \, m = 0.94 \text{ m} \tag{5}$$

Ridge interior (excluding the bottom 10 cm) unit volume (block number by internal layer thickness):

$$v_{int,rdg} = h_c + (h_k - h_c)\frac{1 - \mu_{rbl}}{h_b}(h_b - 2 \cdot 0.1 \text{ m}) = 2.67 \text{ m} \tag{6}$$

Level ice interior (excluding the bottom 10 cm) unit volume:

$$v_{int,fyi/syi} = \left(d_{\frac{fyi}{syi}} - 0.1 \text{ m}\right) = \frac{1.2}{1.3}\text{ m}\left(\frac{\text{FYI}}{\text{SYI}}\right) \tag{7}$$

Estimates for all ice types:

To convert volume estimates to their volumetric fraction, the unit volumes were multiplied by the corresponding ice type areal coverage ($a_{rdg} = 22\%$, $a_{fyi} = 50\%$, $a_{syi} = 28\%$) for the area of about 40 km² around the MOSAiC ice floe from airborne laser scanner surveys. In total, the largest unit volume came from ridge ice (0.82 m), followed by FYI (0.65 m), and SYI interior (0.39 m). To estimate Chl-a and POC concentrations in the different ice types, we used all Chl-a and POC measurements from the ridge, first- and second-year ice taken in July. We assigned to each ice or water type the average Chl-a and POC concentrations (mg m⁻³) to provide an estimate of Chl-a and POC per unit ice area by multiplication by the corresponding total volume fraction (Table 2). Overall, the values used to calculate unit volumes for the different ice types are representative compared to the literature. Block thickness is similar to most ridge observations. Rubble porosity is lower than typical winter porosities close to 30%[4], but similar to lower porosities of 10–20% observed during the melt season[103]. Ridge keel depth is smaller than from drilling surveys[4], but similar to moored sonar observations[3]. The level ice thickness is similar to pan-Arctic satellite estimates[100]. Ridge areal coverage is comparable to moored sonar estimates[3], while the areal fraction of first-year ice is slightly smaller than for the Arctic Ocean currently, with around 70% of first-year ice[104]. The ridge spacing estimated from ICESat−2 laser altimeter for MOSAiC location was around 170 m in March-May 2020, with a pan-Arctic median ridge spacing of 235–280 m[105]. Thus, the conditions during MOSAiC in terms of the fraction of different ice types appear rather representative of current Arctic sea ice conditions. We then combined results from individual ice types to calculate Chl-a and POC concentrations for ridge ice, FYI and SYI (Table 4). Based on the ice type-specific areal coverage, we calculated the contribution of the ice type to the total Chl-a and POC concentration per m² sea ice (Table 4).

## Data availability

The original data used in this study, collected during the MOSAiC expedition, are publicly available and deposited in PANGAEA. All environmental data, including nutrients, POC, biogenic and lithogenic silica, elemental composition of particles, chlorophyll *a*, bacterial production, flow cytometry, and light microscopy counts (simplified) from ridge samples can be found compiled in one data set on PANGAEA ([107]; https://doi.org/10.1594/PANGAEA.983955). Non-ridge data can be found as follows: Dissolved nutrients in the water column can be found in Torres-Valdés et al. ([108]; https://doi.org/10.1594/PANGAEA.966217). Biogenic and lithogenic

**Table 4 | Chlorophyll _a_ (Chl-a) and particulate organic carbon (POC) concentration in different ice types**

| Ice type | Chl-a per volume (mg m$^{-3}$) | Chl-a per area (mg m$^{-2}$) | Chl-a per ice type areal coverage (mg m$^{-2}$) | Relative amount of Chl-a per ice type areal coverage (%) | POC per volume (mg m$^{-3}$) | POC per area (mg m$^{-2}$) | POC per ice type areal coverage (mg m$^{-2}$) | Relative amount of POC per ice type areal coverage (%) |
|---|---|---|---|---|---|---|---|---|
| Ridge | 6 ± 4 | 25 ± 5 | 6 ± 1 | 80 ± 20 | 390 ± 70 | 1570 ± 140 | 350 ± 30 | 41 ± 4 |
| FYI | 1.2 ± 0.3 | 1.5 ± 0.2 | 0.8 ± 0.09 | 11 ± 2 | 340 ± 70 | 440 ± 30 | 220 ± 20 | 26 ± 2 |
| SYI | 1.0 ± 0.2 | 1.4 ± 0.1 | 0.39 ± 0.04 | 6 ± 1 | 730 ± 70 | 1030 ± 90 | 290 ± 30 | 34 ± 3 |

The estimates are based on Chl-a and POC measurements, taking ice type fraction, ice volume, and areal coverage into account. Values are given as mean ± standard error (SE). SE is calculated as the propagation of error.

_FYI_ first-year ice, _SYI_ second-year ice.

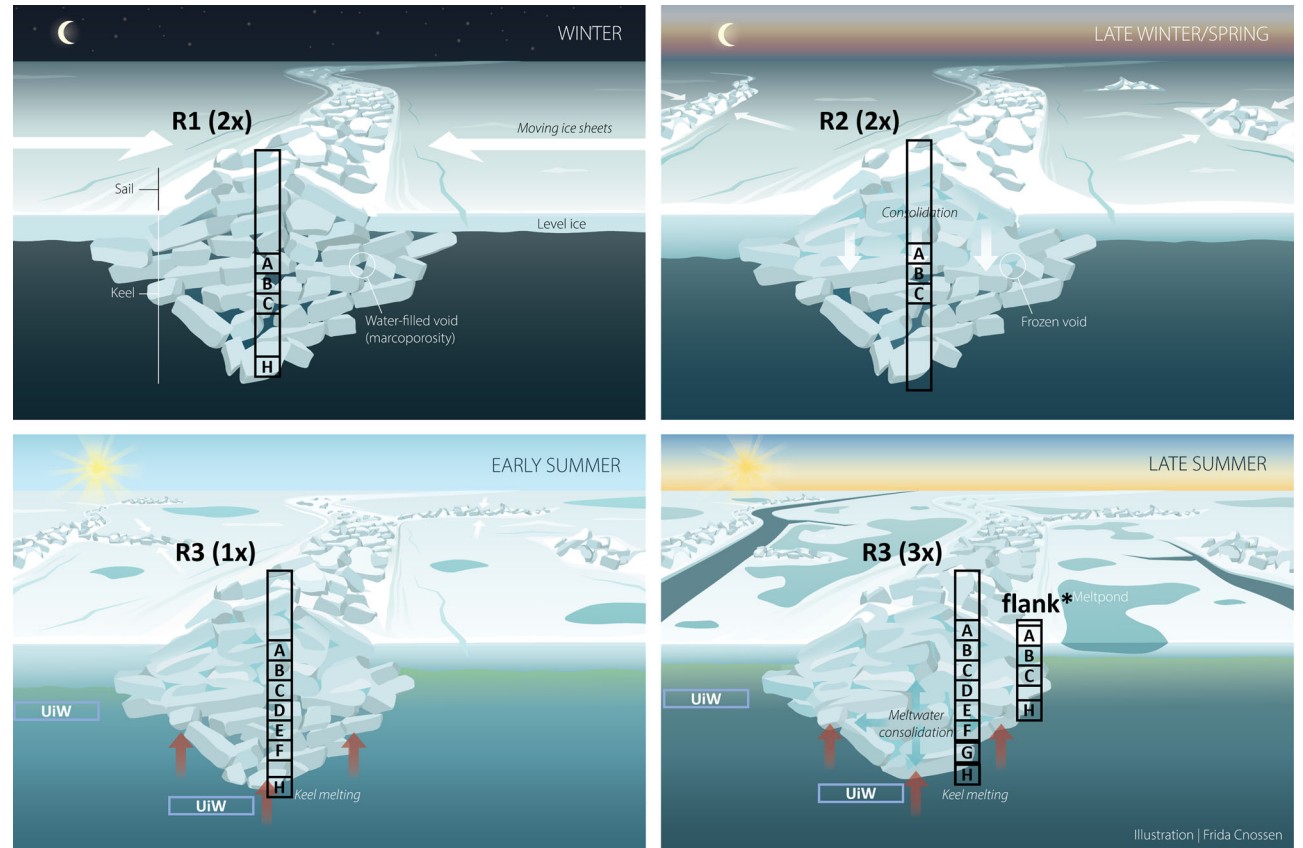

**Fig. 10 | Conceptual visualization of the seasonal evolution of sea ice ridges.** The seasonal evolution of sea ice ridges from winter to summer and relevant processes shaping ridges (e.g., consolidation, melting), as well as which ridge (R1–R3) and how often it was sampled for biological variables during the four represented seasons. The boxes with letters indicate how many and approximately where in the ridge samples were taken (for more details, see Table 3). The white color indicates snow accumulation and the light green color below the ice increase in phytoplankton concentration in the upper water column. UiW under ice water. Credit: Frida Cnossen.

silica in the level ice and water column can be found in Lemke et al. ([109]; https://doi.org/10.1594/PANGAEA.971253). Chlorophyll a concentrations in the level ice and water column can be found in van Leeuwe et al. ([110]; https://doi.org/10.1594/PANGAEA.967448) and Hoppe et al.([111]; https://doi.org/10.1594/PANGAEA.963277). Flow cytometry in the water column and level ice is available in Müller et al. ([112]; https://doi.org/10.1594/PANGAEA.963430) and Müller et al.([113]; https://doi.org/10.1594/PANGAEA.963560). The sequencing raw data are available at the European Nucleotide Archive (ENA), from ridge samples under accession number PRJEB90852 and for non-ridge samples under PRJNA895866. Metagenome data are available under JGI Project 1454842 and Metagenome-assembled genomes (MAGs) recovered from the HAVOC metagenomes under accession number PRJNA1160706. Light microscopy counts (taxonomy) of sea ice (including ridges) can be found in Assmy et al. ([114]; https://doi.org/10.1594/PANGAEA.957637) and for pelagic samples

(incl. ridge voids) in Assmy et al. ([115]; https://doi.org/10.1594/PANGAEA.957640). For R1, drilling data can be found in Salganik et al. ([116]; https://doi.org/10.1594/PANGAEA.960347); coring data in Salganik et al. ([117]; https://doi.org/10.1594/PANGAEA.962542); ice mass balance buoy data in Granskog et al. ([118]; https://doi.org/10.1594/PANGAEA.924269) and in Salganik et al. ([119]; https://doi.org/10.1594/PANGAEA.964023). For R2, the coring data can be found in Salganik et al. ([120], https://doi.org/10.1594/PANGAEA.979884). For R3, drilling data can be found in Salganik et al. ([121]; https://doi.org/10.1594/PANGAEA.953880), and the ice mass balance buoy data in Granskog et al. ([122]; https://doi.org/10.1594/PANGAEA.938354); Level first-year ice coring data can be found in Oggier et al. ([123]; https://doi.org/10.1594/PANGAEA.971385) and level second-year ice in Oggier et al. ([124]; https://doi.org/10.1594/PANGAEA.974764). Multibeam sonar data can be found in Anhaus et al. ([125]; https://doi.org/10.1594/PANGAEA.971872), the airborne laser scanner measurements can be found in Hutter

et al. ([126]; https://doi.org/10.1594/PANGAEA.950896), the helicopter-borne RGB orthomosaics can be found in Neckel et al. ([127]; https://doi.org/10.1594/PANGAEA.949433), and the core hydrographic data can be found in Schulz et al. ([128]; https://doi.org/10.18739/A21J9790B).

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

## Acknowledgements
This work was supported by the Research Council of Norway (grant nos. 280292, 276730, 280531, 328957). P.A. acknowledges support through iC3: Centre for ice, Cryosphere, Carbon and Climate funded by the Research Council of Norway through its Centres of Excellence funding scheme (grant no. 332635). M.A.G. and B.L. acknowledge the support from a fellowship at the Hanse-Wissenschaftskolleg Institute of Advanced Study (Delmenhorst, Germany). A.T. was supported by the Swedish Research Council VR (2018-04685), the Swedish Research Council Formas (2018-00509), and the Swedish Polar Research Secretariat (2019-153), granted to Professor Pauline Snoeijs-Leijonmalm, Stockholm University, Sweden. W.B. was supported by the Natural Environment Research Council and the ARIES Doctoral Training Partnership [grant number NE/S007334/1]. Data used in this manuscript were produced as part of the international Multidisciplinary drifting Observatory for the Study of the Arctic Climate (MOSAiC) with the tag MOSAiC20192020 and the Project_ID: AWI_PS122_00. We thank all people involved in the expedition of the Research Vessel Polarstern during MOSAiC in 2019–2020, as listed in Nixdorf et al.[106]. Specifically, we would like to thank those colleagues for contributing to field sampling and sample analysis, i.e., Katyanne Shoemaker, Steven Fons, Adam Ulfsbo, Laura Heitmann, Kai-Uwe Ludwichowski. We also thank the ROV operators Christian Katlein and Ilkka Matero, as well as Pedro de la Torre, for all technical support with remote instruments.

## Author contributions
Oliver Müller: methodology, formal analysis, investigation, data curation, writing—original draft preparation, visualization, resources, writing—review & editing. Jessie Gardner: methodology, formal analysis, investigation, data curation, resources, writing—review & editing. Lasse Mork Olsen: methodology, investigation, data curation, resources, writing—review & editing. Evgenii Salganik: methodology, formal analysis, investigation, data curation, visualization, resources, writing—review & editing. Philipp Assmy: conceptualization, methodology, resources, data curation, writing—original draft preparation, supervision, writing—review & editing, funding acquisition. Rolf Gradinger: conceptualization, methodology, resources, data curation, writing—original draft preparation, supervision, writing—review & editing, funding acquisition. Gunnar Bratbak: conceptualization, methodology, resources, data curation, writing—original draft preparation, supervision, writing—review & editing, funding acquisition. Clara J.M. Hoppe: methodology, investigation, data curation, resources, writing—review & editing, funding acquisition. Benjamin A. Lange: methodology, investigation, data curation, resources, writing—review & editing. Morven Muilwijk: investigation, resources, writing—review & editing. Dmitry V. Divine: methodology, investigation, resources, writing—review & editing. Nicole Aberle: conceptualization, methodology, resources, data curation, writing—review & editing, funding acquisition. Jeffrey W. Krause: data curation, resources, writing—review & editing. Marit Reigstad: data curation, writing—review & editing, funding acquisition. Eva Leu: conceptualization, resources, data curation, writing—review & editing, funding acquisition. Tatiana M. Tsagaraki: data curation, resources, writing—review & editing. Aud Larsen: conceptualization, methodology, resources, data curation, writing—review & editing, funding acquisition. Knut V. Høyland: conceptualization, methodology, resources, data curation, writing—review & editing, funding acquisition. John Paul Balmonte: methodology, investigation, data curation, writing—review & editing. William Boulton: data curation, resources, writing—review & editing. Håkon Dahle: data curation, writing—review & editing. Lena Eggers: data curation, resources, writing—review & editing. Allison A. Fong: methodology, investigation, writing—review & editing. Gaël Guillou: data curation, writing—review & editing. Benoit Lebreton: data curation, writing—review & editing. Katja Metfies: methodology, investigation, data curation, resources, writing—review & editing. Thomas Mock: data curation, resources, writing—review & editing. Elzbieta Petelenz: data curation, resources, writing—review & editing. Agnieszka Tatarek: methodology, investigation, data curation, resources, writing—review & editing. Sinhué Torres-Valdés: data curation, resources, writing—review & editing. Anders Torstensson: investigation, writing—review & editing. Jozef Wiktor: data curation, resources, writing—review & editing. Mats A. Granskog: conceptualization, methodology, formal analysis, data curation, writing—original draft preparation, writing—review & editing, visualization, supervision, project administration, funding acquisition.

## Funding

## Competing interests
The authors declare no competing interests.

## Additional information

[1]University of Bergen, Department of Biological Sciences, Bergen, Norway. [2]UiT The Arctic University of Norway, Department of Arctic and Marine Biology, Tromsø, Norway. [3]Aqua Kompetanse, Flatanger, Norway. [4]Norwegian Polar Institute, Fram Centre, Tromsø, Norway. [5]Alfred-Wegener-Institut Helmholtz-Zentrum für Polar- und Meeresforschung, Bremerhaven, Germany. [6]Norwegian Geotechnical Institute, Oslo, Norway. [7]Norwegian University of Science and Technology, Department of Biology, Trondheim, Norway. [8]University of Hamburg, Institute of Marine Ecosystem and Fishery Science (IMF), Hamburg, Germany. [9]Stokes School of Marine and Environmental Sciences, University of South Alabama, Mobile, AL, USA. [10]Dauphin Island Sea Lab, Dauphin Island, AL, USA. [11]Akvaplan-niva, Fram Centre, Tromsø, Norway. [12]NORCE Research AS, NORCE Climate & Environment, Bergen, Norway. [13]Norwegian University of Science and Technology, Department of Civil and Environmental Engineering, Trondheim, Norway. [14]Uppsala University, Department of Ecology and Genetics/Limnology, Uppsala, Sweden. [15]Lehigh University, Department of Earth and Environmental Sciences, Bethlehem, PA, USA. [16]University of East Anglia, School of Computing Sciences, Norwich, UK. [17]Joint Research Unit Littoral, Environment and Societies (CNRS—University of La Rochelle), La Rochelle, France. [18]University of East Anglia, School of Environmental Sciences, Norwich, UK. [19]Institute of Oceanology, Polish Academy of Sciences, Sopot, Poland. [20]Swedish University of Agricultural Sciences, Department of Aquatic Sciences and Assessment, Uppsala, Sweden. [21]Swedish Meteorological and Hydrological Institute, Community Planning Services—Oceanography, Västra Frölunda, Sweden. [22]These authors contributed equally: Oliver Müller, Jessie Gardner. ✉e-mail: oliver.muller@uib.no

