## [Transparent Peer Review file · Communications Earth & Environment]

Arctic sea-ice ridges are biomass hotspots harboring diverse microbial communities

Corresponding Author: Dr Oliver Müller

Version 0:

Decision Letter:

Dear Dr Müller,

Please allow us to apologize for sending this late decision on your manuscript titled "Arctic sea-ice ridges: Biomass hotspots harboring diverse microbial communities". It has now been seen by 2 reviewers, and we include their comments at the end of this message. They find your work of interest, but some important points are raised. We are interested in the possibility of publishing your study in Communications Earth & Environment, but would like to consider your responses to these concerns and assess a revised manuscript before we make a final decision on publication.

We therefore invite you to revise and resubmit your manuscript, along with a point-by-point response that takes into account the points raised. Please highlight all changes in the manuscript text file.

Please submit your point-by-point responses as a separate file, distinct from your cover letter where you can add responses to the Editors' comments that you do not want to be made available to the reviewers. Word files are preferred. We recommend that any figures, tables or graphs that are included in the response to reviewers are also included in the main article or Supplementary Information.

Please use the following link to submit your revised manuscript, point-by-point response to the referees' comments (which should be in a separate document to any cover letter), a tracked-changes version of the manuscript (as a PDF file) and the completed checklist:

Link Redacted

We hope to receive your revised paper within six weeks; please let us know if you aren't able to submit it within this time so that we can discuss how best to proceed. If we don't hear from you, and the revision process takes significantly longer, we may close your file. In this event, we will still be happy to reconsider your paper at a later date, as long as nothing similar has been accepted for publication at Communications Earth & Environment or published elsewhere in the meantime.

Please do not hesitate to contact us if you have any questions or would like to discuss these revisions further. We look forward to seeing the revised manuscript and thank you for the opportunity to review your work.

Best regards,

Jose Luis Iriarte Machuca, PhD
Editorial Board Member
Communications Earth & Environment

Mengjie Wang

Associate Editor, Communications Earth & Environment
Consulting Editor, Communications Sustainability
Bluesky: @commsearth.nature.com; @ commssustain.nature.com

EDITORIAL POLICIES AND FORMATTING

- Behavioural and social science
- Ecological, evolutionary & environmental sciences
- Life sciences

Furthermore, please align your manuscript with our format requirements, which are summarized on the following checklist: <https://www.nature.com/documents/commsj-phys-style-formatting-checklist-article.pdf> Communications Earth & Environment formatting checklist

and also in our style and formatting guide <https://www.nature.com/documents/commsj-phys-style-formatting-guide-accept.pdf> Communications Earth & Environment formatting guide .

*** DATA: Communications Earth & Environment endorses the principles of the Enabling FAIR data project (<http://www.copdess.org/enabling-fair-data-project/>). We ask authors to make the data that support their conclusions available in permanent, publically accessible data repositories. (Please contact the editor if you are unable to make your data available).

All Communications Earth & Environment manuscripts must include a section titled "Data Availability" at the end of the Methods section or main text (if no Methods). More information on this policy, is available at <http://www.nature.com/authors/policies/data/data-availability-statements-data-citations.pdf>.

If a community resource is unavailable, data can be submitted to generalist repositories such as <https://figshare.com/> or <http://datadryad.org/> Dryad Digital Repository. Please provide a unique identifier for the data (for example a DOI or a permanent URL) in the data availability statement, if possible. If the repository does not provide identifiers, we encourage authors to supply the search terms that will return the data. For data that have been obtained from publically available sources, please provide a URL and the specific data product name in the data availability statement. Data with a DOI should be further cited in the methods reference section.

REVIEWER COMMENTS:

Reviewer #1 (Remarks to the Author):

This is a very well written manuscript that – for the first time – describes biological and biogeochemical conditions in Arctic pack ice ridge habitats across different seasons (as sampled during the major international MOSAiC drift study). The authors present a unique dataset and comprehensive analyses of the pro- and eukaryotic community composition/biomass/abundance/diversity (analysed using molecular, microscopic and flow cytometric techniques), pigment content, and particulate organic carbon, in combination with other physico-chemical parameters (including particulate elemental composition) in different ridge habitats, as well as in first-year and second-year level sea ice. As such the study

provides many new insights and is highly relevant to the understanding of Arctic sea-ice (micro)-biology and biogeochemistry, and for understanding the role that sea ice ridges may play as potential refugium for sea ice biota under changing ice conditions. The manuscript is well structured, comprises careful and detailed analyses, and all conclusions are clearly derived from the presented data. However - in this reviewer's opinion - there are a few areas that require some further examination and explanation.

Major comments:

1) The authors describe a mistake regarding the handling of temperature-sensitive Chlorophyll a samples (Line 513, Figure S11). These must generally be stored frozen (-80 degC), but were exposed to temperatures above +60 degC for an extended period. The authors run a lab experiment to derive a conversion factor for heat-exposed versus properly handled laboratory/culture samples. This looks fine in general, but to fully convince the reader of the quality of the heat exposed field samples it would be very useful to see the full range of replicate measurements taken for this intercomparison experiment. Please show all 3 replicate data points (as described in the text) in the regression graphs.

2) The authors then use these Chlorophyll data to estimate the relative depth-integrated area; contribution of "ice algal biomass" from ice ridges to an ice area. This is a key message of the manuscript. Given the potential problems with the Chlorophyll samples (as described in 1) it would be great to run this exercise with the particulate organic carbon (POC) data instead - as primary estimate (replacing the current Chlorophyll a calculations) or at least as secondary independent way to estimate and understand the relative contribution of ridges to areal ice algal biomass/ integrated particulate organic carbon (POC). Are Chlorophyll and POC estimates in ridges showing the same trends (relative contributions)? Providing carbon-based estimates will likely also increase the impact/citability of this study/and would provide a baseline for (potential) future modelling studies.

3) Also note: The current Chlorophyll-based estimate for the relative importance of ridges to area-integrated Chlorophyll a are based on July data only. Could a short discussion be added – how the overall estimate might be affected by high biomass in FYI during the spring algal bloom? E.g., do the authors consider their estimate as conservative or not? Related and in addition, the estimates are based on a number on ice ridge physical parameters (all "estimated" averages (Table 3)). Could error bars be considered for these and thus an upper and lower limit of the ice algae Chlorophyll/POC in ridges be calculated, e.g., similar to a sensitivity analysis that provides a range of estimate for the biomass in ridges (their relative contribution to areal ice algal Chla or POC) rather than a single estimate?

4) Void water sampling (Line 473): Void water was sampled with a hand-held bilge pump equipped with a 20 mm hose. Were upper and lower voids sampled consecutively and/or from individual drillings into them? How was cross-contamination from the "interior maze (of often interconnected) water-filled void" avoided? Please provide some additional details on the void water sampling and how cross-contamination has been avoided.

Minor comments:

L36: Could you be more specific? E.g., use "volumetric" ice algal biomass or "area/depth integrated biomass"?

L 88 -96: This is not really an Introduction anymore and could/should be moved to the beginning of Results section.

L 124: Are there any data that show that "R3:..."had consolidated by late winter"?

L 125: Rather than saying "Ice salinity was "2-3 for the full depth"? Please provide a proper mean. This number also contradicts with the range given in this sentence 0.6-5.8 for the available 10 cm sections. Please clarify.

Fig 2: Provide explanation for "Ice with algal inclusion", how is this different from a "frozen void" that likely also has inclusions/algal content?

L 155: "physico-chemical" rather than "physical"?

L183: "in" rather than "from" January?

Figure 3: Figure caption: As different ridges were sampled, I suggest to rephrase "Seasonal changes" to "Illustration of changes in x, xx, xx in samples from different ridges and sampled in different seasons" or similar to clearly state that this is not a comprehensive time-series during which the same habitats were re-sampled over time.

L290: "Superimposed onevent-driven"...? Is ridging and the formation of ridge habitats NOT even-driven as well? Maybe rephrase?

L307 - : Light is a key driver of ice algal biomass development and distribution. The authors provide very clear and detailed discussions on various physico-chemical drivers (salinity, nutrients) ridge habitats and the seasonal changes. Could a short discussion on light levels in ridges be added?

L310: An interesting (potentially citable) study on this topic is also:

Krembs, C., Tuschling, K. & v. Juterzenka, K. The topography of the ice-water interface – its influence on the colonization of sea ice by algae. *Polar Biol* 25, 106–117 (2002). <https://doi.org/10.1007/s003000100318>

L336: "in one of the ridge keels" ?? Was a single ridge keel sampled in Jan and July? If not, I suggest to rephrase.

L402: delete "potentially", word is not needed

Table 2: Could this table be extended to include discrete sampling depth for all listed "Specifics"; also could information be provided how thick the roof, floor, algae inclusion and bottom samples were? Were these all 10-cm cut sections? This latter information is important but somewhat hard to find in the text.

L584: "were" rather the "was" (use plural)

L596: "0" missing

Supplementary Fig 5 a and L27 : "presented as average" or are these median values (shown in the boxplots)?

Overall comments:

This paper provides a thorough spatiotemporal description of the sea ice ridge habitat, which has rarely been described in detail in the past. The claim is made that sea ice ridges provide significant refuge space for overwintering of sea ice algae, along with other microbial community members, and that they contain a high proportion of all algal biomass contributing to primary productivity. A description is provided of the physicochemical setting of sea-ice ridges, and a discussion of biological community composition and seasonal and spatial differentiation is provided. This study does a nice job of laying out the foundation for understanding this complex and understudied ecosystem, but the discussion of the biology feels a bit superficial compared to the physicochemical characterization given the amount of the paper dedicated to these results. Information from 16S and 18S data are primarily used here to differentiate samples, and are not well connected to known functions of the mentioned genera in other sea ice and polar environments. The metagenomic data presented is extremely underutilized in its presentation and adds little to the study as it is. There are some high-level discussions of metabolism that are not complete in their interpretations and may misrepresent the findings. These points could be expanded on and corrected to greatly improve the robustness of the ecological discussion. Given that a major point of this study is that sea-ice ridges harbor the vast majority of primary productivity in the sea ice environment, the discussion of the biogeochemical potential of the inhabitants cannot be skipped when more data is available to inform these points. Overall, this paper is quite well written, the illustrations are excellent, and the description of the sea ice ridge ecosystem is imperative for the field, but the biological discussion needs to be improved to maximize the impact of this study.

Specific comments:

Introduction

L44: This first paragraph would benefit from a more well-rounded introduction to the sea ice ridge environment. As it is, it feels a bit too abrupt into the physical description of ridges before the ecological and biogeochemical importance of this environment has been introduced.

Results

L117: Add unit for salinity, either ppt (‰) or PSU as appropriate throughout

L120 and 125: Specify "bulk ice salinity" for consistency

L206 – 211: I am unsure whether I believe that these are really temporal differences or if they are simply due to the biogeographic heterogeneity between the sampled ridges. Level ice communities can vary within a few meters of distance simply due to spatial heterogeneity. In figure 3, each ridge appears to be quite different overall. I understand that this is likely a combination of time and space and that multi-seasonal sampling of a single ridge was not possible, but I think more reasoning should be added that the environment of each ridge is sufficiently similar to warrant their comparison across seasons. In figure 5, the groupings basically show that R1, R2, and R3 form separate groups and that within R3 the differences are split between liquid and solid phases. This doesn't inherently demonstrate that the difference is seasonal rather than spatial. Do the data actually separate by water/ice temperature, brine volume fraction, PAR, or some other seasonally related environmental variable?

L265 – 280: This is a major underutilization of the metagenomic data. The methods describe 42 metagenomes being sequenced, yet the data describing functional and taxonomic abundances from the metagenomes is not presented in the results. The gene categories discussed here are extremely broad and don't point towards specific functional differences between environmental settings. More functions are listed in figure 7 than are listed in the results, and it isn't obvious how these particular functions differentiate communities when they represent largely general features of prokaryotic metabolism. This data would be better used by associating specific biogeochemically relevant metabolic pathways with the taxa that occupy the different settings of the ridge environment. It would be good to see MAGs reconstructed with this data that can give insight into the capabilities of community members, and to determine whether abundant taxa have meaningful functional relationships or show specific adaptations to the sea ice ridge environment. There are known algal-bacterial association dynamics that could be discussed as well with more detail about potential metabolic exchange. Given the high productivity estimated for the ridges, it would be a major addition to explore the predicted metabolic and biogeochemical outputs by the microbial communities. I recommend digging into this data more deeply to build out the discussion of the functional potential of ridge-associated microbial communities.

Discussion

L344 – 349: This discussion of the function of these taxa is speculative, but it could be informed directly by the metagenomic data. For example, given the high reported relative abundance of *Colwellia*, its functional potential would be extremely informative and would provide some ground truthing for the claim of heterotrophic flexibility and high growth rates.

L356 – 360: This is an oversimplification of gluconeogenesis and metabolic function. Other substrates besides glucose can be used without conducting gluconeogenesis. Many organic acids can be consumed and shunted to the TCA cycle or alternatives without invoking gluconeogenesis, which just is one of many pathways to interconvert metabolites. Gluconeogenesis is a nearly universal metabolism often used for the production of storage molecules and to supplement biosynthesis. Additionally, metabolites exuded from algae are far more diverse than glucose alone, including other sugars, organic acids, and amino acids among more diverse organic molecules, so this logic is flawed in implying that the abundance of these genes represents a shift away from consuming microalgal derived glucose. Rather, the abundance of these genes may imply a shift from active growth to maintenance metabolisms in a more energetically limited frozen environment.

L360 – 362: Algae actively exude metabolites during growth that are available to bacteria not only through degradation, and the exchange is bidirectional with bacteria supplying algae with essential metabolites. Gluconeogenesis is only one of many processes linking this metabolic exchange, so I'm not sure this is conclusive evidence of the activity as presented.

** Visit Nature Portfolio's author and referees' website at www.nature.com/authors for information about policies, services and author benefits**

Communications Earth & Environment is committed to improving transparency in authorship. As part of our efforts in this direction, we are now requesting that all authors identified as 'corresponding author' create and link their Open Researcher and Contributor Identifier (ORCID) with their account on the Manuscript Tracking System prior to acceptance. ORCID helps the scientific community achieve unambiguous attribution of all scholarly contributions. You can create and link your ORCID from the home page of the Manuscript Tracking System by clicking on 'Modify my Springer Nature account' and following the instructions in the link below. Please also inform all co-authors that they can add their ORCIDs to their accounts and that they must do so prior to acceptance.

If you experience problems in linking your ORCID, please contact the Platform Support Helpdesk.

Version 1:

Decision Letter:

Dear Dr Müller,

Your manuscript titled "Arctic sea-ice ridges: Biomass hotspots harboring diverse microbial communities" has now been seen by our reviewers, whose comments appear below. In light of their advice we are delighted to say that we are happy, in principle, to publish a suitably revised version in Communications Earth & Environment.

We therefore invite you to revise your paper one last time to address the remaining concerns of our reviewers. At the same time we ask that you edit your manuscript to comply with our format requirements and to maximise the accessibility and therefore the impact of your work.

EDITORIAL REQUESTS:

****Please take care to match our formatting and policy requirements. We will check revised manuscript and return manuscripts that do not comply. Such requests will lead to delays. ****

SUBMISSION INFORMATION:

OPEN ACCESS:

Communications Earth & Environment is a fully open access journal. Articles are made freely accessible on publication. For further information about article processing charges, open access funding, and advice and support from Nature Portfolio, please visit <https://www.nature.com/commsenv/open-access>

At acceptance, you will be provided with instructions for completing the open access licence agreement on behalf of all

authors. This grants us the necessary permissions to publish your paper. Additionally, you will be asked to declare that all required third party permissions have been obtained, and to provide billing information in order to pay the article-processing charge (APC).

Link Redacted

Best regards,

Jose Luis Iriarte Machuca, PhD
Editorial Board Member
Communications Earth & Environment

Mengjie Wang
Associate Editor, Communications Earth & Environment
Consulting Editor, Communications Sustainability
Bluesky: @commsearth.nature.com; @commssustain.nature.com

REVIEWERS' COMMENTS:

Reviewer #1 (Remarks to the Author):

Review of the revised manuscript (R1) of Müller et al. "Arctic sea-ice ridges: Biomass hotspots harboring diverse microbial communities"

Thank you for your detailed response and consideration of my and the Reviewer 2 comments. The manuscript has gotten a bit long but has much improved and now includes a (more) comprehensive data evaluation of genomics data. I am also satisfied with your explanation of the handling of the heated Chlorophyll samples and think that the data can be trusted. Thank you also for including the POC data analyses to estimate the contribution of biomass in ridges to overall areal biomass.

I have only very minor suggestions to this version:

Line 210: can you provide Spearman's rho here as well?

Line 304/305: Clarify sentence: HNA bacteria ...were?

Line 358: Seasonal changes.... at different timescales. This is not clear to me as seasonal changes have seasonal timescales – suggest to rephrase.

Line424/426: Suggest to rephrase this sentence: "Finally,, adds" does not sound correct to me

L530: "RV Polarstern"? rather than just Polarstern?

Line 647 and 668: Check figure numbers given in these lines. Should "Fig. 12" in line 647 be "Fig. 11"?

Suppl. L87: Figure 11a

Suppl. L123: Paulino et al. is duplicated

Suppl. L136: brackets missing around (1999)

Reviewer #2 (Remarks to the Author):

The revised results and discussion arising from the addition of metabolic information from the MAGs has made an excellent addition to this study. The metabolic landscape of the microbial communities inhabiting the sea ice ridge environment is now much more clear and illustrates a dynamic cycle between primary and secondary productivity across seasons that helps to explain the ecological impact of this previously understudied environment. This revision involved a significant addition that was well executed and adds a lot of strength to the manuscript. This study represents a significant and fundamental addition to our knowledge of sea ice ecosystems.

Reviewer #3 (Remarks to the Author):

I commend the authors on this comprehensive manuscript describing the physical setting, microbial ecology, and biogeochemistry of sea ice ridges in the Arctic. Ridges have long intrigued sea ice scientists, but the numerous logistical difficulties inherent in sampling them have prevented extensive research into their properties, despite the large fraction of total Arctic sea ice (and potentially productivity) they encompass. The well-designed research program described here provides some of the most detailed measurements to date on the sea ice ridge habitat, and offers conceptual and modeling frameworks that will increase the viability of future ice ridge research. This is important because the Arctic pack ice is changing, and will continue to change as warming continues, so ridges may become even more important refugia for Arctic sea ice biota as this century progresses. I have no major concerns with the results as reported here and look forward to the eventual publication of this manuscript.

** Visit Nature Portfolio's author and referees' website at www.nature.com/authors for information about policies, services and author benefits**

REVIEWER COMMENTS:

Reviewer #1 (Remarks to the Author):

This is a very well written manuscript that – for the first time – describes biological and biogeochemical conditions in Arctic pack ice ridge habitats across different seasons (as sampled during the major international MOSAiC drift study). The authors present a unique dataset and comprehensive analyses of the pro- and eukaryotic community composition/biomass/abundance/diversity (analysed using molecular, microscopic and flow cytometric techniques), pigment content, and particulate organic carbon, in combination with other physico-chemical parameters (including particulate elemental composition) in different ridge habitats, as well as in first-year and second-year level sea ice. As such the study provides many new insights and is highly relevant to the understanding of Arctic sea-ice (micro)-biology and biogeochemistry, and for understanding the role that sea ice ridges may play as potential refugium for sea ice biota under changing ice conditions. The manuscript is well structured, comprises careful and detailed analyses, and all conclusions are clearly derived from the presented data. However - in this reviewer's opinion - there are a few areas that require some further examination and explanation.

Thank you for your positive evaluation of our work, and constructive suggestions for improvement.

Major comments:

1) The authors describe a mistake regarding the handling of temperature-sensitive Chlorophyll a samples (Line 513, Figure S11). These must generally be stored frozen (-80 degC) , but were exposed to temperatures above +60 degC for an extended period. The authors run a lab experiment to derive a conversion factor for heat-exposed versus properly handled laboratory/culture samples. This looks fine in general, but to fully convince the reader of the quality of the heat exposed field samples it would be very useful to see the full range of replicate measurements taken for this intercomparison experiment. Please show all 3 replicate data points (as described in the text) in the regression graphs.

We agree that this is an important point, and now represent all replicate points visually by using x and y error bars accounting for the replication of both treatments. We had to reduce the point size to the minimum size in order to visualize the error-bars.

2) The authors then use these Chlorophyll data to estimate the relative depth-integrated area; contribution of “ice algal biomass” from ice ridges to an ice area. This is a key message of the manuscript. Given the potential problems with the Chlorophyll samples (as described in 1) it would be great to run this exercise with the particulate organic carbon (POC) data instead - as primary estimate (replacing the current Chlorophyll a calculations) or at least as secondary independent way to estimate and understand the relative contribution of ridges to areal ice algal biomass/ integrated particulate organic carbon (POC). Are Chlorophyll and POC estimates in ridges showing the same trends (relative contributions)? Providing carbon-based estimates will likely also increase the impact/citability of this study/and would provide a baseline for (potential) future modelling studies.

Thank you for the suggestion of providing estimates of POC concentrations in the different ice types in addition to the Chl-a estimates. We agree that this is a valuable addition and useful for future modelling work, and at the same time provide arguments why Chl-a might be the better indicator of

in situ algal biomass in sea ice, since POC values (in particular the SYI) seem to be affected by organic material, incorporated from external sources (benthic/terrestrial) and carried with the ice. Performing the same analysis using POC data as for the Chl-a data we found that the relative distribution of POC per sea ice area was 41 ± 4 , 26 ± 2 , 34 ± 3 (% \pm SE) for ridge, FYI and SYI respectively. The C:Chl-a ratio in the ridge, FYI and SYI was 31, 148 and 432 respectively, suggesting that incorporated organic material and transported within the ice (see discussion and Krumpen et al. 2020) was likely a significant fraction of the organic carbon in SYI. Using POC as a biomass estimate in sea ice may hence be more controversial than using Chl-a.

We added the new calculations for POC in Tables 3 and 4 as well as text parts:

Results: (Lines 176-178 in the track-changes document): “In level FYI, the highest POC concentrations were measured in the bottom ice sections in July (up to 1590 mg C m^{-3}). The overall highest POC values were measured in SYI and were found in the interior ice sections at 80 to 100 cm depth varying between 1540 and 3300 mg C m^{-3} .” and (Lines 184-185): “Likewise, we find that the ridges ($1570 \pm 140 \text{ mg C m}^{-2}$) contain more POC than FYI ($440 \pm 30 \text{ mg C m}^{-2}$) and SYI ($1030 \pm 90 \text{ mg C m}^{-2}$).”

Discussion (Lines 416-426 in the track-changes document): “The calculated POC standing stocks in the different ice types also show that the relative amount of POC per ice type areal coverage was higher in the ridge than in FYI and SYI. Surprisingly, the concentration of POC in SYI was much higher in the interior than in the bottom layer (Table 3), suggesting that a significant fraction of the carbon in SYI is likely stored material. This observation is supported by the C:Chl-a ratio, which was highest in SYI (432), compared to much lower values in FYI (148) and the ridge (31). These ratios indicate that ridges provide more habitable space supporting active algal growth, while level ice may contain relatively more stored organic material from previous growing seasons and depending on the origin and age of the ice also material incorporated where the ice formed, such as sediment flora and fauna from the Siberian Shelf (Fig. S9⁴⁴). Finally, since non-phototrophic organisms (such as heterotrophic nanoflagellates and bacteria) additionally contribute to POC, adds to our interpretation that Chl-a may be a better indicator of actively growing *in situ* algal biomass in sea ice compared to POC.”

3) Also note: The current Chlorophyll-based estimate for the relative importance of ridges to area-integrated Chlorophyll a are based on July data only. Could a short discussion be added – how the overall estimate might be affected by high biomass in FYI during the spring algal bloom? E.g., do the authors consider their estimate as conservative or not?

This is an interesting point and we did therefore some additional calculations where we used the highest Chl-a concentration (7.8 mg m^{-3}) we observed in FYI during the MOSAiC campaign in the bottom ice (0-10cm) fraction, i.e. twice the average concentration from July. Using this value in our calculations, the relative contribution of FYI Chl-a per sea ice area would only increase from 11% to 14%. In addition, the ridges we studied were formed prior to the spring bloom so the high Chl-a biomass in the ridges has likely been produced *in situ* and does not originate from spring ice algal bloom biomass incorporated with FYI into the ridge. During the spring bloom the biomass in both FYI and the ridges was most certainly higher than what we measured in this study (i.e. July). Their relative contribution to Chl-a per sea ice area may however still be about the same, and in this sense, we may consider our estimates as conservative. Nevertheless, we need more field observations to address this issue more accurately.

We added a section in the discussion where we discuss the raised topic (Lines 408-414 in the track-changes document): “This estimate is based on July measurements only and is hence only a snapshot from that season. During the spring ice algal bloom, the biomass in the bottom ice of both level ice and within the ridges was most certainly higher than what we measured in this study, but their relative contribution to Chl-a per sea ice area may however still be comparable, due to the larger volume of habitable space in ridges. It is important to note that the ridges we studied were formed prior to the ice algal spring bloom. Therefore, the high Chl-a biomass suggests in situ growth rather than biomass introduced with the incorporation of level ice as the main contributor to biomass build up in ridges.”

Related and in addition, the estimates are based on a number on ice ridge physical parameters (all “estimated” averages (Table 3)). Could error bars be considered for these and thus an upper and lower limit of the ice algae Chlorophyll/POC in ridges be calculated, e.g., similar to a sensitivity analysis that provides a range of estimate for the biomass in ridges (their relative contribution to areal ice algal Chl-a or POC) rather than a single estimate?

Chlorophyll concentrations have been recalculated (and corrected) and are now reported with standard error (SE) for all measurements. Chlorophyll per area ice type in the Arctic is based on concentration, ice type volume and ice type areal coverage (Table 3). Calculation of chlorophyll concentrations per volume, area and areal coverage of ridge Ice, FYI and SYI take ice type fraction, volume and areal coverage into account and SE is calculated accordingly as propagation of error (Table 4).

4) Void water sampling (Line 473): Void water was sampled with a hand-held bilge pump equipped with a 20 mm hose. Were upper and lower voids sampled consecutively and/or from individual drillings into them? How was cross-contamination from the “interior maze (of often interconnected) water-filled void” avoided? Please provide some additional details on the void water sampling and how cross-contamination has been avoided.

This is indeed an important consideration and at the same time a difficult to assess completely. In general, the reviewer is correct in pointing out that cross-contamination can't be avoided as voids are interconnected, however our measurements of salinity indicate different water origins for voids that we sampled; the upper (salinity of 8) and the lower void (salinity of 30), which were 1 m apart. Hence, we are very confident that there was no vertical cross-contamination, but horizontally there is the possibility to have sampled water from other connecting void at the same horizon. We added following statement to the method section (Lines 598-602 in the track-changes document): “When two different void locations were sampled during the same sampling event (R3: 03.07.2020), due to complexity and interconnectivity of voids we cannot exclude that the sampled water was only from one individual void and potentially a mix from neighboring, connected voids. However, due to large

vertical distance (1 m) between the upper and lower void we can exclude vertical cross-contamination.”

Minor comments:

L36: Could you be more specific? E.g., use “volumetric” ice algal biomass or “area/depth integrated biomass”?

Thank you for pointing this out, we added as suggested to inform the reader that it is area integrated biomass.

L 88 -96: This is not really an Introduction anymore and could/should be moved to the beginning of Results section.

This paragraph was designed to fulfil the guide for authors guidelines for CommsEnv, that states that; “The final paragraph [of the Introduction] should be a brief summary of the major results and conclusions.”. Thus we prefer to keep it as is as we attempted to follow the journal guideline. (see <https://www.nature.com/documents/commsj-phys-style-formatting-guide-accept.pdf>)

L 124: Are there any data that show that “R3:...”had consolidated by late winter”?

This is specified in the method section (Line 558 and corresponding references in the track-changes document) and with reference to Salganik et al., 2023 (Observations of preferential summer melt of Arctic sea-ice ridge keels from repeated multibeam sonar surveys) where it is written: <Jaridge [named R3 in this paper] was formed between 4 and 12 February 2020, based on the visual inspection of sea-ice surface elevation models from an airborne laser scanner (Jutila et al., 2023).>

L 125: Rather than saying “Ice salinity was “2-3 for the full depth””? Please provide a proper mean. This number also contradicts with the range given in this sentence 0.6-5.8 for the available 10 cm sections. Please clarify.

We changed as suggested, now providing the mean (2.1 ± 1.2). The minimum and maximum values from salinity measurements that cover the entire ridge are in alignment (0.1 – 6.5) with salinity measurements of the 10 cm ice sections that were chosen for sampling of biological parameters (0.6-5.8).

Fig 2: Provide explanation for “Ice with algal inclusion”, how is this different from a “frozen void” that likely also has inclusions/algal content?

We added the term “visible” to clarify the difference. While it is correct that frozen voids also incorporated algal material it was not visible during the sampling.

L 155: “physico-chemical” rather than “physical”?

Yes, this has been changed accordingly.

L183: “in” rather than “from” January?

Yes, this has been changed accordingly.

Figure 3: Figure caption: As different ridges were sampled, I suggest to rephrase “Seasonal changes” to “Illustration of changes in x, xx, xx in samples from different ridges and sampled in different seasons” or similar to clearly state that this is not a comprehensive time-series during which the same habitats were re-sampled over time.

While we indicated the different ridges/seasons on top of the panels, we acknowledge that the general impression from the figure could be continuous re-sampling which was not intended and hence agree that it is better to be more specific and made the changes as suggested.

L290: “Superimposed onevent-driven”...? Is ridging and the formation of ridge habitats NOT even-driven as well? Maybe rephrase?

We had internal discussion concerning this exact phrase, and agreed that we wanted to include “event-driven” to mark the meltwater driven consolidation of ridges in summer as it is strictly speaking not just a seasonal change. It is however also correct that the formation of ridges is also event-driven. We therefore changed to the following: “Superimposed on seasonal changes in the Arctic icescape at different timescales, the complex three-dimensional structure of ridges, combined with strong vertical and horizontal environmental gradients, facilitates large habitat diversity on small spatial scales.”

L307 - : Light is a key driver of ice algal biomass development and distribution. The authors provide very clear and detailed discussions on various physico-chemical drivers (salinity, nutrients) ridge habitats and the seasonal changes. Could a short discussion on light levels in ridges be added?

Thank you for this suggestion. Indeed light is also a key driver. However, there are to our knowledge no direct measurements of light level within ridges, and only some anecdotal evidence of the complexity of the light field within ridges (and thus seasonality), due to the complex structure of the ice blocks in the ridges. We added the following now L378-380 (in the track-changes document): “A conceptual modelling study⁴⁰ and supported by recent field observations⁴¹ suggest that ridges can be a conduit for light and thus affect light availability for primary producers.”

40. Katlein, C. *et al.* The Three-Dimensional Light Field Within Sea Ice Ridges. *Geophys. Res. Lett.* **48**, e2021GL093207 (2021).

41. Castellani, G. *et al.* Arctic sea-ice ridges: a major contributor to algal habitable space in spring. *Front. Mar. Sci.* **12**, 1653882 (2025).

L310: An interesting (potentially citable) study on this topic is also:

Krembs, C., Tuschling, K. & v. Juterzenka, K. The topography of the ice-water interface – its influence on the colonization of sea ice by algae. *Polar Biol* 25, 106–117

(2002). <https://doi.org/10.1007/s003000100318>

Thank you for this suggestion. While this is indeed an interesting paper, we do not necessarily think it has direct implications for our work with ridges, given the “bumps” in their experiment scale better to the underside of level ice, and not the much larger (orders of magnitude) ridge keels, therefore we aren’t convinced this is a relevant citation to add.

L336: “in one of the ridge keels” ?? Was a single ridge keel sampled in Jan and July? If not, I suggest to rephrase.

Yes, this was not clear based on the chosen wording and we changed accordingly.

L402: delete “potentially”, word is not needed

This has been changed accordingly.

Table 2: Could this table be extended to include discrete sampling depth for all listed “Specifics”; also could information be provided how thick the roof, floor, algae inclusion and bottom samples were? Were these all 10-cm cut sections? This latter information is important but somewhat hard to find in the text.

We agree that it is useful to have this information easily available and not just in the published datafile on PANGAEA and included it as suggested.

L584: “were” rather the “was” (use plural)

This has been changed accordingly.

L596: “0” missing

The version was correct, but a bracket was missing and has been corrected now.

Supplementary Fig 5 a and L27 : “presented as average” or are these median values (shown in the boxplots)?

Thank you for pointing this out, it is indeed the median values and has been changed accordingly.

Reviewer #2 (Remarks to the Author):

Overall comments:

This paper provides a thorough spatiotemporal description of the sea ice ridge habitat, which has rarely been described in detail in the past. The claim is made that sea ice ridges provide significant refuge space for overwintering of sea ice algae, along with other microbial community members, and that they contain a high proportion of all algal biomass contributing to primary productivity. A

description is provided of the physicochemical setting of sea-ice ridges, and a discussion of biological community composition and seasonal and spatial differentiation is provided. This study does a nice job of laying out the foundation for understanding this complex and understudied ecosystem, but the discussion of the biology feels a bit superficial compared to the physicochemical characterization given the amount of the paper dedicated to these results. Information from 16S and 18S data are primarily used here to differentiate samples, and are not well connected to known functions of the mentioned genera in other sea ice and polar environments. The metagenomic data presented is extremely underutilized in its presentation and adds little to the study as it is. There are some high-level discussions of metabolism that are not complete in their interpretations and may misrepresent the findings. These points could be expanded on and corrected to greatly improve the robustness of the ecological discussion. Given that a major point of this study is that sea-ice ridges harbor the vast majority of primary productivity in the sea ice environment, the discussion of the biogeochemical potential of the inhabitants cannot be skipped when more data is available to inform these points. Overall, this paper is quite well written, the illustrations are excellent, and the description of the sea ice ridge ecosystem is imperative for the field, but the biological discussion needs to be improved to maximize the impact of this study.

Thank you for your positive evaluation of our work, and constructive suggestions for improvement.

Specific comments:

Introduction

L44: This first paragraph would benefit from a more well-rounded introduction to the sea ice ridge environment. As it is, it feels a bit too abrupt into the physical description of ridges before the ecological and biogeochemical importance of this environment has been introduced.

We shortened the first section by combining the last two sentences and hope to improve thereby the overall flow. We still think it is warranted to first point out the widespread presence of sea ice ridges in the Arctic, and their physical formation/appearance in the ice pack that makes them potentially disproportionately important for the sea-ice ecosystem. We think this is suitable, especially considering the wide-readership of the journal.

We changed sentence at the end of this section (L48-49 in the track-changes document) to reduce this section and improve the flow: "The keel is composed of randomly arranged ice blocks (rubble), with voids initially filled with seawater (typically with a fraction of 30%)⁷."

Results

L117: Add unit for salinity, either ppt (‰) or PSU as appropriate throughout

We report salinity as practical salinity, on the Practical Salinity Scale (PSS-78), and by definition this is dimensionless because it is a conductivity ratio (note that this was mentioned in the Methods in the original submission, lines 582-583 in the revised track-changes document). Thus using "PSU" as a unit is wrong, and misused by many authors, because there is no such "unit". It is correct to report practical salinity without any units. See e.g. <https://salinometry.com/pss-78/>

To clarify this, we have now written with the first use of salinity in the Results part of the main text (Line 119-120 in the revised track-changes document) the same information in brief, and continue to give unitless values of salinity in the text. This should help the reader.

L120 and 125: Specify “bulk ice salinity” for consistency

Thank you for pointing this out, this has been changed accordingly.

L206 – 211: I am unsure whether I believe that these are really temporal differences or if they are simply due to the biogeographic heterogeneity between the sampled ridges. Level ice communities can vary within a few meters of distance simply due to spatial heterogeneity. In figure 3, each ridge appears to be quite different overall. I understand that this is likely a combination of time and space and that multi-seasonal sampling of a single ridge was not possible, but I think more reasoning should be added that the environment of each ridge is sufficiently similar to warrant their comparison across seasons. In figure 5, the groupings basically show that R1, R2, and R3 form separate groups and that within R3 the differences are split between liquid and solid phases. This doesn't inherently demonstrate that the difference is seasonal rather than spatial. Do the data actually separate by water/ice temperature, brine volume fraction, PAR, or some other seasonally related environmental variable?

It is true that spatial heterogeneity in sea ice is an important factor, and in fact is something that we found to be a strong factor in the sampled ridge in July. However, the presented differences of community composition between summer and winter/spring are to our understanding strongly linked to seasonality. The community differences illustrated in Fig. 5 are indeed very similar to the sample grouping when using environmental parameters only (Fig.2 – PCA). We see the same grouping based on brine salinity, brine volume and temperature into winter (R1; R2 was excluded due to missing nutrient data) and summer (R3) and the separation of R3 based on frozen or water-filled voids. This is very much what we also observe for the grouping in Fig. 5. We think that this is sufficient presented and described but do however agree that a larger number of samples, especially from the winter/spring period would provide a better picture of spatial heterogeneity on top of the seen temporal differences.

L265 – 280: This is a major underutilization of the metagenomic data. The methods describe 42 metagenomes being sequenced, yet the data describing functional and taxonomic abundances from the metagenomes is not presented in the results. The gene categories discussed here are extremely broad and don't point towards specific functional differences between environmental settings. More functions are listed in figure 7 than are listed in the results, and it isn't obvious how these particular functions differentiate communities when they represent largely general features of prokaryotic metabolism. This data would be better used by associating specific biogeochemically relevant metabolic pathways with the taxa that occupy the different settings of the ridge environment. It would be good to see MAGs reconstructed with this data that can give insight into the capabilities of community members, and to determine whether abundant taxa have meaningful functional relationships or show specific adaptations to the sea ice ridge environment. There are known algal-bacterial association dynamics that could be discussed as well with more detail about potential metabolic exchange. Given the high productivity estimated for the ridges, it would be a major addition to explore the predicted metabolic and biogeochemical outputs by the microbial

communities. I recommend digging into this data more deeply to build out the discussion of the functional potential of ridge-associated microbial communities.

Thank you for raising this concern regarding the non-used potential of the metagenomic data. We completely agree that we only present a minor fraction of the metagenomic data and at a very broad level. We appreciate in particular the suggestion of including information from the reconstructed MAGs, since they have not been included so far. Accordingly, we used the dbCAN and METABOLIC software to perform an extensive annotation on proteins (.faa files) derived from 932 MAGs that have been reconstructed previously from HAVOC metagenomic data (described in Boulton et al., 2025; PRJNA1160706). In these analyses we were particularly interested in exploring consistencies between variations in gene distributions versus variations in the taxonomic community structures and biological data across sample types. The results are described in the revised manuscript and represent substantial additions to the result (L309-330 and a new Fig 8; in the track-changes document) and discussion sections (L442-451; L460-467; L476-484 in the track-changes document). Indeed, we find that these additions strengthen the paper and addresses the reviewers concerns and recommendations regarding the lack of detailed analyses of the specific functional differences between environmental settings and metabolic capabilities of detected taxa. It would have been possible to perform even more extensive analyses of detected MAGs, but we feel that a more thorough analyses would fit into a separate metagenomic-focused manuscript which we hope to write in the future.

Discussion

L344 – 349: This discussion of the function of these taxa is speculative, but it could be informed directly by the metagenomic data. For example, given the high reported relative abundance of *Colwellia*, its functional potential would be extremely informative and would provide some ground truthing for the claim of heterotrophic flexibility and high growth rates.

We agree that this has mainly been based on information from literature and we added evidence from the analysis of the MAGs in more general terms in L442-451 (in the track-changes document) and in particular for *Colwellia* based on one of the 12 MAGs taxonomically assigned to *Colwellia* in L460-467 (in the track-changes document).

L356 – 360: This is an oversimplification of gluconeogenesis and metabolic function. Other substrates besides glucose can be used without conducting gluconeogenesis. Many organic acids can be consumed and shunted to the TCA cycle or alternatives without invoking gluconeogenesis, which just is one of many pathways to interconvert metabolites. Gluconeogenesis is a nearly universal metabolism often used for the production of storage molecules and to supplement biosynthesis. Additionally, metabolites exuded from algae are far more diverse than glucose alone, including other sugars, organic acids, and amino acids among more diverse organic molecules, so this logic is flawed in implying that the abundance of these genes represents a shift away from consuming microalgal derived glucose. Rather, the abundance of these genes may imply a shift from active growth to maintenance metabolisms in a more energetically limited frozen environment.

We thank the reviewer for the detailed insights showing the rather large complexity of this topic that we certainly were not able to capture in our discussion. We therefore decided to replace this paragraph with more concrete information from the analysis of the MAGs with focus on the

differences in carbohydrate-active enzymes between the ridge ice samples when voids were water-filled or frozen (new Fig 8b and L322-330 and L447-451 in the track-changes document).

L360 – 362: Algae actively exude metabolites during growth that are available to bacteria not only through degradation, and the exchange is bidirectional with bacteria supplying algae with essential metabolites. Gluconeogenesis is only one of many processes linking this metabolic exchange, so I'm not sure this is conclusive evidence of the activity as presented.

We agree that we oversimplified here and therefore removed this paragraph as described earlier. To acknowledge the bi-directional exchange and known algal-bacterial association dynamics we also included a section in the results where we present specifically differences in abundance of genes encoding for enzymes involved in algae-growth promoting cofactor pathways (L299-302 in the track-changes document) and in more general terms other metabolic functions (L317-320 in the track-changes document). This is then taken up in the discussion, with particular focus on carbon degradation processes L442-451, L460-467 and L476-484 since this is relevant to provide the metabolic context to the observed changes in terms of increase in bacterial abundance and production as response to freezing of the water-filled voids.

Dear reviewers,

We want to thank all reviewers for very helpful comments and suggestions and are pleased to hear that our last revisions and responses improved the manuscript.

All last comments from reviewer 1 have been addressed, please see the details below:

Sincerely,

Oliver Müller, on behalf of all the co-authors

Line 210: can you provide Spearman's rho here as well?

It has been included.

Line 304/305: Clarify sentence: HNA bacteria ...were?

This is correct and has been changed.

Line 358: Seasonal changes.... at different timescales. This is not clear to me as seasonal changes have seasonal timescales – suggest to rephrase.

We removed “at different timescales” to avoid confusion and to be more clear.

Line 424/426: Suggest to rephrase this sentence: “Finally,, adds” does not sound correct to me

This sentence has been rephrased: “In addition, non-phototrophic organisms (such as heterotrophic nanoflagellates and bacteria) also contribute to the POC pool, further corroborating our interpretation that Chl-a may be a better indicator of actively growing *in situ* algal biomass in sea ice compared to POC.”

L530: “RV Polarstern”? rather than just Polarstern?

This has been changed accordingly.

Line 647 and 668: Check figure numbers given in these lines. Should “Fig. 12” in line 647 be “Fig. 11”?

Thank you for pointing this out. Fig. 12 in line 647 is correct, but we changed “Fig. 12” in line 668 to Fig. 13.

Suppl. L87: Figure 11a

Thank you for pointing this out, we corrected this section and changed to Supplementary Fig. 12.

Suppl. L123: Paulino et al. is duplicated

This has been changed accordingly.

Suppl. L136: brackets missing around (1999)

This has been changed accordingly.